# Revisiting Injective Attacks on Recommender Systems

**Haoyang LI**[1], **Shimin DI**[1],[*] **Lei CHEN**[2]

[1]Computer Science and Engineering, HKUST, Hong Kong, China
[2]Data Science and Analytics, HKUST(GZ), Guangzhou, China
`{hlicg,sdiaa,leichen}@cse.ust.hk`

## Abstract

Recent studies have demonstrated that recommender systems (RecSys) are vulnerable to injective attacks. Given a limited fake user budget, attackers can inject fake users with carefully designed behaviors into the open platforms, making RecSys recommend a target item to more real users for profits. In this paper, we first revisit existing attackers and reveal that they suffer from the difficulty-agnostic and diversity-deficit issues. Existing attackers concentrate their efforts on difficult users who have low tendencies toward the target item, thus reducing their effectiveness. Moreover, they are incapable of affecting the target RecSys to recommend the target item to real users in a diverse manner, because their generated fake user behaviors are dominated by large communities. To alleviate these two issues, we propose a difficulty and diversity aware attacker, namely DADA. We design the difficulty-aware and diversity-aware objectives to enable easy users from various communities to contribute more weights when optimizing attackers. By incorporating these two objectives, the proposed attacker DADA can concentrate on easy users while also affecting a broader range of real users simultaneously, thereby boosting the effectiveness. Extensive experiments on three real-world datasets demonstrate the effectiveness of our proposed attacker.

## 1 introduction

In the era of big data, recommender systems (RecSys) [8, 28, 36, 39, 53] are a key component to help users locate items that they may be interested in. Generally, given the user behaviors, such as item purchase history and movie viewing history, RecSys learn the latent tendency of users regarding each item, and then recommend the top-N matched items for each user. Over the past decade, various RecSys have been proposed to provide services for users. Recent studies [17, 24, 33, 45, 51] demonstrate that RecSys are vulnerable to adversarial attacks, e.g., fake user behaviors. It is because the training of RecSys relies on user behaviors and attackers can easily inject fake users into open-world web platforms. Specifically, attackers want to promote an item to real users, which can increase the opportunity that users purchase this target item and thus bring economic profits. To achieve such goals, attackers generally inject a limited number of fake users with designed behaviors, i.e., *injective attacks*. Then, the target RecSys may be affected with these injected fake users, and then recommends the target item into the top-N item list of real users.

Based on the techniques that are employed to design the behaviors of injected fake users, existing attackers [4, 6, 10, 11, 12, 20, 30, 38, 41, 45, 49, 51, 52] can be classified into three types: *heuristics-based* [4, 30, 49], *neural network-based* [10, 38, 45, 52], and *gradient-based* attackers [6, 11, 12, 20, 41, 51]. Heuristics-based attackers [4, 30, 49] sample popular items together with the target item as the behaviors of each fake user to increase the co-occurrence between popular items and the target item, i.e., trying to increase the popularity of the target item. Then, RecSys can be affected,

---

[*]Corresponding author

36th Conference on Neural Information Processing Systems (NeurIPS 2022).

and will recommend the target item to more users. However, such a way does not directly optimize the attack objective, reducing the effectiveness of attackers [41, 52]. Instead, neural network-based attackers [10, 38, 45, 52] utilize neural networks to generate influential fake user behaviors. They maximize the attack objective by optimizing the parameters of neural networks. Gradient-based attackers [6, 11, 12, 20, 41, 51] adopt the continuous relaxation to project user behaviors from the discrete space into the continuous space. This enables attackers to parameterize the fake user behaviors and maximize the attack objective by optimizing them with gradients.

Though existing attackers can achieve reasonably good attacking performance, they still suffer from two crucial issues from the local and global views: *difficulty-agnostic* and *diversity-deficit*. First, existing attackers do not consider the promotion difficulty of each user regarding the target item. Particularly, users who have a higher (resp. lower) tendency toward the target item are referred to easier (resp. more difficult) users. According to the formulated attack objective of existing attackers, difficult users will contribute more weights to optimize attackers' parameters, e.g., neural network parameters for generating fake users. It makes attackers put more efforts to recommend the target item to difficult users. However, the RecSys platforms will usually monitor the user registration, i.e., attackers only have a very limited budget for generating fake users. Thus, attackers that focus on difficult users may only achieve small attacking gain, i.e., the target item can only be recommended to a small number of users. Second, the generated fake behaviors of existing attackers cannot influence real users diversely. Specifically, real users with different preferences are dispersed into different communities. Communities with more users may dominate the optimization of attackers. Consequently, the generated fake users cannot affect RecSys to promote the target item to various communities, decreasing the effectiveness of attackers.

To address *difficulty-agnostic* and *diversity-deficit* issues, we propose a Difficulty-Aware and Diversity-Aware attacker, namely DADA, which takes the promotion difficulty and diversity into consideration simultaneously. First, to handle the difficulty-agnostic issue, we propose a novel difficulty-aware objective that more emphasizes easier users than difficult users when optimizing attack parameters. Moreover, we propose a diversity-aware attack objective that allows the generated fake user behaviors to affect the target RecSys to recommend the target item to real users diversely, thereby avoiding the diversity-deficit issue. By incorporating both attack objectives, our proposed attacker DADA can concentrate on easy users and generate fake user behaviors that are capable of affecting real users in a diverse manner. Overall, our contributions are summarized as follows:

- We revisit attackers on RecSys and reveal that existing attackers suffer from *difficulty-agnostic* and *diversity-deficit* issues. More specifically, the objective designs in existing attackers make them more focus on promoting the target item to those difficult users in the dominant community. However, it is hard to handle difficult users with only a limited number of generated fake users. Besides, the generated fake users tend to fall into the dominant community, which is not helpful to affect real users in other communities.

- We propose two novel attack objectives (difficulty-aware and diversity-aware objectives) that force the RecSys attacker to attack easy users distributed in various communities. Generally, the proposed objectives could enable the easy users and unaffected real users to contribute more weights when optimizing the parameters of the RecSys attacker.

- Experiments on three real-world datasets demonstrate that the proposed attacker DADA achieves superior performance compared with state-of-the-art attackers. Besides, extensive experiments show that the generated fake users indeed focus on easy users in a diverse way.

## 2 Related Work

Formally, let $U$ and $I$ denote users and items, respectively. $\mathbf{R} \in \{0, 1\}^{|U| \times |I|}$ denotes the records of user-item interactions. Specifically, each entry $\mathbf{R}[u][i]$ denotes whether the user $u \in U$ interacts with the item $i \in I$, and $I_u = \{i : \mathbf{R}[u][i] = 1\}$ denotes the interacted item set of each user $u \in U$.

### 2.1 Recommender System

Recently, various RecSys [47, 48] have been proposed and deployed in the website applications, which recommend the items for users that match the preference of users. Formally, given users $U$, items $I$, and user behaviors $\mathbf{R} \in \{0, 1\}^{|U| \times |I|}$, RecSys $\mathcal{M}_\theta$ learns the latent tendency score

Table 1: Summary of existing RecSys attackers. ✓ and × indicate that the attacker can and cannot satisfy the property, respectively. $t$ is the target item that attackers are interested in. The function $f(\mathbf{R}[u][i], \mathbf{R}[u][t]) = 1$ if both $\mathbf{R}[u][i] = 1$ and $\mathbf{R}[u][t] = 1$, and $g(x) = \frac{1}{1+exp(-x/b)}$. $b$, $b_1$, and $b_2$ are hyperparameters, and $a$ and $c$ are constants. $\mathbf{S}'$ and $\mathbf{S}$ are learned tendency before and after attacks, respectively. We discuss them in detail in Sec. 2.2.

| Type | Attacker | Attack Objective | | |
|---|---|---|---|---|
| | | $\mathcal{O}_{atk}(\cdot)$ Formulation | Difficulty | Diversity |
| Heuristic-based | Popular [29] | - | × | × |
| | CoVis [49] | $\sum_{i\in I} \mathbb{I}(\sum_{u\in U} f(\mathbf{R}[u][i], \mathbf{R}[u][t])) \geq a)$ | × | × |
| Neural network-based | PoisonRec [38] | $\frac{1}{|U^r|}\sum_{u\in U^r} \mathbb{I}(t, \Gamma_u)$ | × | × |
| | CopyAttack [10] | $\frac{1}{|U^r|}\sum_{u\in U^r} \mathbb{I}(t, \Gamma_u)$ | × | × |
| | TrialAttack [45] | $-\sum_{u\in U^r}\sum_{i\in\Gamma_u} \ln(g(\mathbf{S}[u][i] - \mathbf{S}[u][t]))$ | × | × |
| Gradient-based | PGA [20] | $\sum_{u\in U^r} \mathbf{S}[u][t]$ | × | × |
| | SGLD [20] | $\sum_{u\in U^r} \mathbf{S}[u][t]$ | × | × |
| | SRWA [12] | $\sum_{u\in U^r} \mathbf{S}[u][t]$ | × | × |
| | TNA [11] | $-\sum_{u\in U^r}\sum_{i\in\Gamma_u} g(\mathbf{S}[u][i] - \mathbf{S}[u][t])$ | × | × |
| | PAPU [51] | $\sum_{u\in U^r}\sum_{i\in\Gamma_u} g(\mathbf{S}[u][t] - \mathbf{S}[u][i])$ | × | × |
| | RevAdv [41] | $\sum_{u\in U^r} \log(\frac{exp(\mathbf{S}[u][t])}{\sum_{i\in I} exp(\mathbf{S}[u][i])})$ | × | × |
| | DADA (Ours) | $\sum_{u\in U^r \wedge\, t\notin\Gamma_u} -\log(1 - b_1\cdot\mathbf{S}[u][t])$ $+\lambda\cdot\sigma(b_2*(\mathbf{S}[u][t] - \mathbf{S}'[u][t] - c))$ | ✓ | ✓ |

$\mathbf{S} = \mathcal{M}_\theta(\mathbf{R}) \in \mathbb{R}^{|U|\times|I|}$ between users and items. The parameters $\theta$ can be optimized as follows:

$$\theta^* = \arg\min_\theta \mathcal{L}_{tra}(\mathcal{M}_\theta(\mathbf{R}), \mathbf{R}), \tag{1}$$

where $\mathcal{L}_{tra}(\mathcal{M}_{\theta^*}(\mathbf{R}), \mathbf{R})$ measures the training loss of RecSys model $\mathcal{M}_\theta$ based on latent tendency $\mathbf{S}$ and user behaviors $\mathbf{R}$. Then, based on the learned tendency $\mathbf{S}$, the RecSys $\mathcal{M}_\theta$ recommends the top-N un-interacted items $\Gamma_u$ with the highest tendency scores for each user $u \in U$, i.e., $\Gamma_u \subset I \setminus I_u$ and $|\Gamma_u| = N$. Generally, current RecSys can be classified into three categories, i.e., neighborhood-based [27, 35], matrix-factorization-based [18, 32], and neural-network-based [15, 50]. Take the fundamental matrix-factorization RecSys WRMF [18] as an example, it maps users $U$ and items $I$ into latent representations $\mathbf{U} \in \mathbb{R}^{|U|\times d_x}$ and $\mathbf{I} \in \mathbb{R}^{|I|\times d_x}$, respectively. Then, it predicts the tendency $\mathbf{S} = \mathbf{U}\mathbf{I}^T$ between users and items. Particularly, WRMF takes $\theta = \{\mathbf{U}, \mathbf{I}\}$ as the parameter and reformulates Eq. (1) into $\arg\min_{\theta=\{\mathbf{U},\mathbf{I}\}} \sum_{u\in U, i\in I} w_{ui}(\mathbf{S}[u][i] - \mathbf{R}[u][i])^2 + \eta(\|\mathbf{U}\|_2^2 + \|\mathbf{I}\|_2^2)$. where $\|\cdot\|_2$ is the $l_2$ norm distance, $w_{ui}$ is the weight hyperparamter to differentiate the existing and un-existing interactions, and $\eta$ is a regularization hyperparameter. More details of recommender systems can be referred to comprehensive surveys [47, 48].

## 2.2 Injective Attacks on Recommender System

Given a target item that attackers are interested in, attackers can inject fake users with carefully crafted behaviors into platforms. The attacker's goal is to affect the RecSys to promote the target item into the top-N recommendation item list of as many real users as possible. Note that attackers can also demote a target item, i.e., maximizing the number of real users whose top-N recommendation list does not include the target item. In this paper, we focus on promotion, because attacks for demotion can be constructed by a similar technique as promotion [10, 12, 38, 49, 52].

Formally, let $U^r$ and $U^f$ denote real users and fake users, respectively. Given the real user behaviors $\mathbf{R}^r \in \{0,1\}^{|U^r|\times|I|}$, the target item $t$, and the target RecSys model $\mathcal{M}_\theta$, the goal is to generate the fake user data $\mathbf{R}^f \in \{0,1\}^{|U^f|\times|I|}$ with the constraints, injection budget $|U^f| \leq \delta$ and the maximum number $\|\mathbf{R}^f[u]\|_0 \leq \tau$ of items that each fake user can interact with, which can maximize the attack objective as follows:

$$\max_{\mathbf{R}^f} \mathcal{O}_{atk}(\mathcal{M}_{\theta^*}(\mathbf{R}); U^r, t), \tag{2}$$

$$s.t. \quad \theta^* = \arg\min_\theta \mathcal{L}_{tra}(\mathcal{M}_\theta(\mathbf{R}), \mathbf{R}), |U^f| \leq \delta, \|\mathbf{R}^f[u]\|_0 \leq \tau \text{ for any } u \in U^f,$$

where $\mathcal{O}_{atk}(\cdot)$ is the attack objective, $\mathcal{L}_{tra}(\cdot)$ is the training loss of RecSys $\mathcal{M}_\theta$ as Eq. (1), and $\mathbf{R} = \begin{bmatrix} \mathbf{R}^r \\ \mathbf{R}^f \end{bmatrix} \in \{0,1\}^{(|U^r|+|U^f|)\times|I|}$ is matrix concatenation. Based on this formulation, various attackers against RecSys are mainly different from three aspects: (1). The formulation of the attack objective $\mathcal{O}_{atk}$; (2). Technique used to optimize Eq. (2); (3). Knowledge accessed by attackers.

**Formulation of $\mathcal{O}_{atk}$.** Hit ratio [10, 38] is a common formulation, i.e., $\mathcal{O}_{atk}(\cdot) = \frac{1}{|U^r|}\sum_{u \in U^r} \mathbb{I}(t, \Gamma_u)$, where indicator $\mathbb{I}(t, \Gamma_u) = 1$ if $t$ is recommended in the top-N item list of $u$, i.e., $t \in \Gamma_u$. Also, several attackers [11, 12, 20, 41, 45, 51, 52] try to maximize the latent tendency between real users and the target item, and propose several attack objective variants. The objective details of these attackers are listed in Tab. 1. For clarification, we specifically list the part of the attack objective that is related to the learned latent tendency $\mathbf{S}$ between real users $U^r$ and items $i \in I$, and omit unrelated parts, such as the regularization of attacker neural network parameters. Note that several attackers [11, 41, 51] try to minimize the attack loss $\mathcal{L}_{atk}$. For generality, we set their attack objective $\mathcal{O}_{atk} = -\mathcal{L}_{atk}$ and use two notations interchangeably in this paper.

**Technique.** It is a non-trivial task to generate the exact optimal fake user behaviors $\mathbf{R}^f$ that can achieve the maximum attack objective due to the exponential candidate search space, i.e., $O(|I|^{\tau \cdot |U^f|})$. As shown in Tab. 1, regarding the technique that generates behaviors for injected fake users, there are three types of attackers: *heuristics-based*, *neural network-based*, and *gradient-based* attackers. As introduced in Sec. 1, heuristics-based attackers [4, 30, 49] cannot cover various behavior patterns and do not directly optimize the attack objective, leading to unsatisfied performance [52, 41]. Instead, neural network-based attackers [10, 38, 45, 52] propose neural networks $\mathcal{A}_\varphi$ parameterized with $\varphi$ to learn probability distributions of selected items for each fake user. Then, they sample fake user behaviors $\mathbf{R}^f \sim \mathcal{A}_\varphi(\mathbf{R}^r)$ accordingly. Formally, the parameter $\varphi$ can be optimized by maximizing the attack objective as $\max_{\varphi, \theta} E_{\mathbf{R}^f \sim \mathcal{A}_\varphi(\mathbf{R}^r)}[\mathcal{O}_{atk}(\mathcal{M}_{\theta^*}(\mathbf{R}); U^r, t)]$. Besides, gradient-based attackers $\mathcal{A}$ [6, 11, 12, 20, 41, 51] relax the discrete fake user behaviors into continuous $\tilde{\mathbf{R}}^f \in \mathbb{R}^{|U^f| \times |I|}$. Then, they directly parameterize $\tilde{\mathbf{R}}^f$ and maximize attack objective as $\max_{\tilde{\mathbf{R}}^f, \theta} \mathcal{O}_{atk}(\mathcal{M}_{\theta^*}(\tilde{\mathbf{R}}); U^r, t)$. After finishing the optimization of $\tilde{\mathbf{R}}^f$, they use a projector function to project the continuous user behaviors $\tilde{\mathbf{R}}^f$ into discrete fake behaviors $\mathbf{R}^f$.

In this paper, we design a gradient-based attacker to generate fake user behaviors. For generality, we use $\varphi$ to denote the parameterized $\tilde{\mathbf{R}}^f$. Specifically, existing gradient-based attackers $\mathcal{A}$ use gradient ascent approach [34] to optimize $\varphi$ by maximizing the attack objective $\mathcal{O}_{atk}(\cdot)$ as follows

$$\varphi \leftarrow \varphi + \alpha \cdot \sum_{u \in U^r} \frac{\partial \mathcal{O}_{atk}(\mathcal{M}_{\theta^*}(\mathbf{R}); u, t)}{\partial \varphi}, \tag{3}$$

where $\alpha$ is the learning rate, and $\mathcal{O}_{atk}(\mathcal{M}_{\theta^*}(\mathbf{R}); u, t)$ denotes the objective value of user $u$ regarding the target item $t$. Particularly, when a user $u$ provides more gradients $\frac{\partial \mathcal{O}_{atk}(\mathcal{M}_{\theta^*}(\mathbf{R}); u, t)}{\partial \varphi}$ to optimize parameter $\varphi$, the attacker will prioritize maximizing the user's objective. For simplicity, we use $\mathcal{O}_{atk}(U^r, t)$ and $\mathcal{O}_{atk}(u, t)$ to denote $\mathcal{O}_{atk}(\mathcal{M}_{\theta^*}(\mathbf{R}); U^r, t)$ and $\mathcal{O}_{atk}(\mathcal{M}_{\theta^*}(\mathbf{R}); u, t)$, respectively.

**Knowledge.** Regarding the knowledge accessed by attackers, existing attackers can be classified into two categories. White-box attackers [11, 12, 20] take real user behaviors and the target RecSys parameters as inputs to generate fake user behaviors. Unlike white-box attackers, gray-box attackers [10, 38, 41, 45, 51, 52] assume that the target RecSys parameters are inaccessible. Instead, the first type of gray-box attackers [10, 38] assume that the top-N item lists of real users recommended by the target RecSys are available. Also, the second type of gray-box attackers [41, 45, 51, 52] only access user behaviors and use a surrogate RecSys $\hat{\mathcal{M}}_{\hat{\theta}}$ to replace the target RecSys $\mathcal{M}_\theta$ in Eq. (2), which is expected to mimic the target RecSys. In summary, the second type of gray-box attackers are more realistic than white-box and the first type of gray-box. The reason is that the target RecSys parameters and the recommendation lists of real users are difficult to obtain. Instead, due to the openness of web platforms, it is easy to collect real user behaviors [41, 51], such as movie viewing history. More discussions on user behavior collection can be referred to Appx. B.5. Thus, we focus on the second type of gray-box attacks on RecSys in this paper.

## 3 DADA Attacker

In this section, we introduce our attacker DADA. We first propose two attack objectives that consider the promotion difficulty and the diversity of users, respectively. Then, we formulate a novel difficulty and diversity-aware attack objective that enables our attacker DADA to simultaneously focus on easy users and affect real users diversely, and propose an efficient greedy-based algorithm to optimize it.

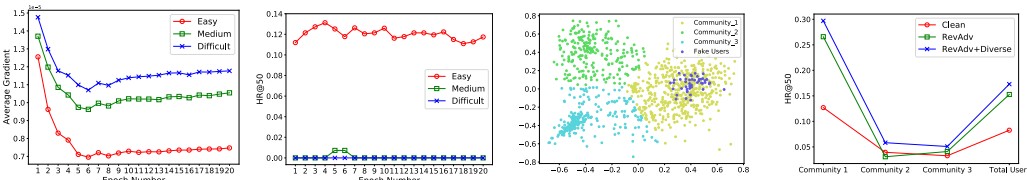

(a) Average gradient on each user group.  (b) HR@50 on each user group.  (c) Fake user distribution on each community.  (d) HR@50 on each community.

Figure 1: Difficulty-agnostic and diversity-deficit analysis.

## 3.1 Difficulty-aware Attack Objective

### 3.1.1 Motivational Observations of Difficulty-agnostic

As discussed in Sec. 1, given a target item and a set of real users, existing RecSys attackers more focus on promoting the target item to difficult users when designing fake user behaviors. We first demonstrate several motivational observations in existing works. Specifically, we employ the representative and state-of-the-art gradient-based attackers, including RevAdv [41], TNA [11], and PAPU [51], to inject fake users into the ML-100K dataset [14], where fake user budget $\delta = 0.01|U^r|$. Specifically, we use the representative RecSys WRMF [18] as the target RecSys $\mathcal{M}_\theta$ for evaluation. We first train WRMF on ML-100K without attacks, and output the latent tendency $\mathbf{S}$ between each user $u \in U^r$ and each item $i \in I$. We then randomly select an item $t \in I$ as the target item. For simplicity and clarification, we separate real users into three groups as an example. Specifically, if the latent tendency of a user $u$ toward the target item $\mathbf{S}[u][t]$ is at the top of $\{33.3\%, 33.3\% \sim 66.7\%, 66.7\% \sim 100\%\}$ largest among all items $I$, it will be regarded as $\{easy, medium, difficult\}$ users, respectively. Then, we can obtain three groups $\{U^e, U^m, U^d\}$.

For clarification, we show the result of RevAdv in Fig. 1 and similar results of TNA and PAPU in Appx. A.1.1. For each group $U^g \in \{U^e, U^m, U^d\}$, we show its average gradient for optimizing the attacker parameter $\varphi$ in Fig. 1 (a), i.e., $\frac{\sum_{u \in U^g} \|\nabla_\varphi \mathcal{L}_{atk}(u,t)\|_1}{|U^g| \cdot (|U^f| \cdot |I|)}$. We can easily observe that the gradient value is directly proportional to the degree of difficulty. In other words, the more difficult it is, the greater gradient it is. It indicates that RevAdv puts more efforts to promote the target item to difficult and medium users than easy users. Then, we investigate the hit ratio of 50 (HR@50) performance on three different user groups. As illustrated in Fig. 1 (b), despite that RevAdv puts more effort on medium and difficult users, the HR@50 of them is only marginally improved. Moreover, putting more effort on medium and difficult users leads that the HR@50 of easy users increases unstably. Therefore, with a limited budget for fake users, it is preferable to concentrate on easy users, as it is easier to promote the target item for them.

### 3.1.2 Insights on Difficulty-agnostic Issue

Motivated by the observations, we plan to design a new difficulty-aware attack objective, which enables attackers to focus on easy users primarily and achieve more recommendation gain. The basic idea is to let easier users outweigh difficult users when optimizing the attacker's parameter $\varphi$. As introduced in Eq. (3), the gradients for optimizing attack parameter $\varphi$ that each user provides are related to the attack objective $\mathcal{O}_{atk}$. Particularly, due to the chain rule, the gradients $\frac{\partial \mathcal{O}_{atk}(u,t)}{\partial \varphi}$ of each user $u \in U^r$ can be formulated as $\frac{\partial \mathcal{O}_{atk}(u,t)}{\partial \varphi} = \frac{\partial \mathcal{O}_{atk}(u,t)}{\partial \mathbf{S}[u][t]} \cdot \frac{\partial \mathbf{S}[u][t]}{\partial \varphi}$, where $\frac{\partial \mathcal{O}_{atk}(u,t)}{\partial \mathbf{S}[u][t]}$ only relates to the attacker objective design. $\frac{\partial \mathbf{S}[u][t]}{\partial \varphi}$ only relies on the attacker parameters, which is fixed and unrelated to attack objective $\mathcal{O}_{atk}(\cdot)$. Thus, we can increase the $\frac{\partial \mathcal{O}_{atk}(u,t)}{\partial \mathbf{S}[u][t]}$ of easier user to let them more determine the optimization direction of attacker parameters. Besides, if a user has received the target item in the top-N recommendation list, it indicates that the RecSys has promoted the target item to this user successfully. Thus, this user should not provide any gradient, because increasing its tendency toward the target item will not have any improvement on the recommendation gain.

Based on the above analysis, the difficulty-aware attack objective should satisfy two *difficulty properties*: (1) given two users $u$ and $v$, if the tendency score $\mathbf{S}[v][t] < \mathbf{S}[u][t]$ and $t \notin \Gamma_u$, $\frac{\partial \mathcal{O}_{atk}(v,t)}{\partial \mathbf{S}[v][t]} < \frac{\partial \mathcal{O}_{atk}(u,t)}{\partial \mathbf{S}[u][t]}$. (2) if the target item $t \in \Gamma_u$, $\frac{\partial \mathcal{O}_{atk}(u,t)}{\partial \mathbf{S}[u][t]} = 0$. As shown in Tab. 1, the attack objectives of existing attackers do not satisfy both properties simultaneously. In this paper, based on the two properties, we formulate a difficulty-aware attack objective. Formally, given the tendency $\hat{\mathbf{S}}$

learned by the surrogate RecSys $\hat{\mathcal{M}}_{\hat{\theta}}$, the target item $t$, the difficulty-aware attack objective regarding each real user $u \in U^r$ is formulated as follows:

$$Dict(\hat{\mathcal{M}}_{\hat{\theta}^*}(\mathbf{R}); u, t) = \begin{cases} -\log(1 - b_1 \cdot \hat{\mathbf{S}}[u][t]), \ t \notin \Gamma_u, \\ 0, \qquad\qquad\qquad\ \ t \in \Gamma_u \end{cases}, \tag{4}$$

where $b_1$ is a positive hyperparameter. Furthermore, we use the following Proposition 1 to show that Eq. (4) enables easier users to more dominate the the attack parameter optimization than difficult users. Due to space limit, we put the proof in Appx. A.2.1.

**Proposition 1.** *Our difficulty-aware attack objective satisfies difficulty property 1 and property 2.*

### 3.2 Diversity-aware Attack Objective

#### 3.2.1 Motivational Observations of Diversity-deficit

As introduced in Sec. 1, existing attackers have the *diversity-deficit* issue that the generated fake behaviors are dominated by large communities, where users in each community have similar preferences. Specifically, as introduced in Eq. (3), each real user $u \in U^r$ will provide gradients to optimize the attackers' parameters $\varphi$. Since similar real users have similar preferences on items, real users can form several communities $\{C_j\}_{j=1}^{n_c}$, where each community $C_j$ has $n_j$ real users. Thus, from the community perspective, the optimization process of the attacker parameter $\varphi$ via maximizing the attack objective $\mathcal{O}_{atk}(\cdot)$ can be reformulated as follows: $\varphi = \varphi + \alpha \cdot \sum_{j=1}^{n_c} \sum_{u \in C_j} \frac{\partial \mathcal{O}_{atk}(\mathcal{M}_{\theta^*}(\mathbf{R}); u, t)}{\partial \varphi}$. Intuitively, users in the same community who are similar will give the gradients in similar directions. Thus, there is a potential risk that the large communities $C_k$ will dominate the optimization direction of attacker parameter $\varphi$ via $\sum_{u \in C_k} \frac{\partial \mathcal{O}_{atk}(\mathcal{M}_{\theta^*}(\mathbf{R}); u, t)}{\partial \varphi}$. Consequently, this community will dominate the generated fake user behaviors. The generated fake user behaviors can only manipulate the target RecSys to promote the target item to the dominated communities. In another words, the target RecSys is incapable of promoting the target item to more users in other communities diversely.

Similar to Sec. 3.1.1, we demonstrate several motivational observations. We use state-of-the-art gradient-based attackers RevAdv [41], TNA [11], and PAPU [51], and neural network-based attacker TrialAttack [45], to generate $0.05|U^r|$ fake users and inject them into the ML-100K dataset [14]. Then, we train the target RecSys WRMF and study the attacking results. For simplicity and clarification, we put the result of RevAdv in Fig. 1 and similar results of TNA, PAPU, and TrialAttack in Appx. A.1.2. Following [41], we extract the representations of real and fake users learned by WRMF, and use PCA [1] to visualize them. Based on the result of PCA, we separate real users into three communities by KMeans [19]. As shown in Fig. 1 (c), the fake users (blue points) tend to form a cluster that is distributed in the community 1 (yellow points). It indicates that community 1 dominates the generated fake users. It is because users in the same community will generate gradients with similar directions. Thus, large communities with more users are more likely to dominate the optimization directions of the attacker parameter. Consequently, RecSys can only be affected by the generated fake user behaviors to recommend the target item to more users in the community 1.

To further verify this, we list the hit ratio 50 (HR@50) of each community before and after injecting fake users in Fig. 1 (d). It shows that the generated fake users improve the HR@50 of the community 1 significantly, while maintaining similar HR@50 in the other two communities. To enable RevAdv to affect users diversely, we add our formulated diversity-aware attack objective (see Sec. 3.2.2) into RevAdv's objective. As shown in Fig. 1 (d), RevAdv+Diverse improves HR@50 of community 2 and 3. Particularly, HR@50 on community 1 increases as well. As described in Sec. 3.1.1, RevAdv more focuses on difficult users, thus the generated fake user behaviors may not be able to affect the target RecSys to recommend the target item to other less difficult users in community 1. Fortunately, the diverse-aware objective allows RevAdv to put more efforts on these users, thereby improving the performance of community 1.

#### 3.2.2 Insights on Diversity-deficit Issue

Motivated by these observations, we propose a diversity-aware attack objective that enables fake user behaviors to affect real users diversely. The basic idea is that if the real users are less affected by fake user behavior, the diversity-aware objective will concentrate on these real users in the subsequent gradient computation process. Then, the attacker will be optimized to generate fake user behaviors that can affect these users. In such a way, users in different communities can be affected diversely.

Specifically, the influence of fake user behaviors on each user regarding the target item can be measured by the difference between tendencies before and after attacks. Formally, let $\mathbf{S}'$ and $\mathbf{S}$ denote the tendencies learned by RecSys $\mathcal{M}$ before and after attacks, the influence of fake user behaviors $\mathbf{R}^f$ on user $u$ regarding the target item $t$ is computed by $\triangle\mathbf{S}[u][t] = \mathbf{S}[u][t] - \mathbf{S}'[u][t]$. To enable attackers to affect more real users, the attack objective should focus on users with a smaller $\triangle\mathbf{S}[u][t]$ when optimizing the attacker parameter $\varphi$. Specifically, according to the chain rule and $\frac{\partial\triangle\mathbf{S}[u][t]}{\partial\mathbf{S}[u][t]} = 1$, the gradients $\frac{\partial\mathcal{O}_{atk}(u,t)}{\partial\varphi}$ regarding the attacker parameter $\varphi$ in Eq. (3) of each user $u \in U^r$ can be formulated as $\frac{\partial\mathcal{O}_{atk}(u,t)}{\partial\varphi} = \frac{\partial\mathcal{O}_{atk}(u,t)}{\partial\triangle\mathbf{S}[u][t]} \cdot \frac{\partial\mathbf{S}[u][t]}{\partial\varphi}$. Thus, we can increase the $\frac{\partial\mathcal{O}_{atk}(u,t)}{\partial\triangle\mathbf{S}[u][t]}$ of real users with smaller influence $\triangle\mathbf{S}[u][t]$ to let them more determine the attacker parameter optimization direction. Similar to Sec. 3.1.2, the diversity-aware attacker objective should satisfy two *diversity properties*: (1) given two users $u$ and $v$, if $\triangle\mathbf{S}[v][t] > \triangle\mathbf{S}[u][t]$, and $t \notin \Gamma_u$, $\frac{\partial\mathcal{O}_{atk}(v,t)}{\partial\triangle\mathbf{S}[v][t]} < \frac{\partial\mathcal{O}_{atk}(u,t)}{\partial\triangle\mathbf{S}[u][t]}$; (2) if the target item $t$ has been recommended to a user $u$, the user $u$ should have no effect on the attackers' parameter optimization, i.e., if $t \in \Gamma_u$, $\frac{\partial\mathcal{O}_{atk}(u,t)}{\partial\triangle\mathbf{S}[u][t]} = 0$.

Obviously, existing attackers do not satisfy both properties because they do not consider the difference of tendency between before and after attacks. Based on the two properties, we formulate a diversity-aware attack objective. Formally, given the target item $t$, and the real user $u \in U^r$, let $\hat{\mathbf{S}}'$ and $\hat{\mathbf{S}}$ denote the tendencies learned by the surrogate RecSys $\hat{\mathcal{M}}_{\hat{\theta}}$ before and after injecting fake user behaviors $\mathbf{R}^f$, the diversity-aware objective on each user $u \in U^r$ regarding the target item $t$ is computed as:

$$Div(\hat{\mathbf{S}}[u][t]; u, t, \hat{\mathbf{S}}'[u][t]) = \begin{cases} \sigma(b_2 * (\hat{\mathbf{S}}[u][t] - \hat{\mathbf{S}}'[u][t] - c)), & t \notin \Gamma_u \\ 0, & t \in \Gamma_u \end{cases}, \quad (5)$$

where $c = \min_{v \in U^r}(\hat{\mathbf{S}}[v][t] - \hat{\mathbf{S}}'[v][t])$ is a constant for normalization, and $b_2$ is a positive hyperparameter to scale the tendency difference. $\sigma(x) = \frac{1}{1+\exp(-x)}$ is the Sigmoid function. Additionally, we use Proposition 2 to show that Eq. (5) enables users with a smaller $\triangle\hat{\mathbf{S}}[u][t]$ to contribute more weights to the attack parameter optimization. Due to space limit, we put the proof in Appx. A.2.2.

**Proposition 2.** *Our diversity-aware attack objective satisfies diversity property 1 and property 2.*

### 3.3 Difficulty and Diversity Aware Attacker

#### 3.3.1 Problem Formulation

Based on two measurements proposed in Eq. (4) and Eq. (5), we formally formulate the Difficulty-Aware and Diversity-Aware (DADA) RecSys attacking problem as follows.

**Definition 1** (DADA Problem). *Given the real user behaviors $\mathbf{R}^r \in \{0,1\}^{|U^r| \times |I|}$, the target item $t$, and the surrogate RecSys $\hat{\mathcal{M}}_{\hat{\theta}}$, the goal is to generate the fake user behaviors $\mathbf{R}^f \in \{0,1\}^{|U^f| \times |I|}$ with the constraints, injection budget $|U^f| \le \delta$ and the maximum number $\|\mathbf{R}^f[u]\|_0 \le \tau$ of items that each fake user can interact with, which can maximize the attack objective as follows:*

$$\max_{\mathbf{R}^f} \sum_{u \in U^r} Dict(\hat{\mathcal{M}}_{\hat{\theta}^*}(\mathbf{R}); u, t) + \lambda \cdot Div(\hat{\mathbf{S}}[u][t]; u, t, \hat{\mathbf{S}}'[u][t]) \quad (6)$$

$$s.t. \quad \hat{\theta}^* = \arg\min_{\hat{\theta}} \mathcal{L}_{tra}(\hat{\mathcal{M}}_{\hat{\theta}}(\mathbf{R}), \mathbf{R}), |U^f| \le \delta, \|\mathbf{R}^f[u]\|_0 \le \tau \text{ for any } u \in U^f,$$

*where $\lambda$ is a trade-off hyperparamter, $\mathbf{R} \in \{0,1\}^{(|U^r|+|U^f|) \times |I|}$ concatenates the real and fake user representations, $\hat{\mathbf{S}}$ and $\hat{\mathbf{S}}'$ are the learned latent tendencies of the surrogate RecSys $\hat{\mathcal{M}}_{\hat{\theta}}$ on $\mathbf{R}$ and $\mathbf{R}^r$, respectively. $\mathcal{L}_{tra}(\cdot)$ is the training loss of $\hat{\mathcal{M}}_{\hat{\theta}}$ as in Eq. (1).*

#### 3.3.2 Greedy Algorithm

Due to the exponential search space $O(|I|^{\tau \cdot |U^f|})$, it is nontrivial to find the exact optimal fake user behaviors for the difficulty-aware and diversity-aware optimization problem in Eq (6). Thus, we propose a gradient-based greedy algorithm to maximize the objective in Alg. 1 in Appx. A.3. With the help of gradients, our algorithm can give a good direction to find effective fake user behaviors. Specifically, we first initialize the fake user behaviors $\mathbf{R}^f \in \{0,1\}^{|U^f| \times |I|}$ (line 1). In each epoch $l$, we first train the surrogate model $\hat{\mathcal{M}}_{\hat{\theta}}$ on $\mathbf{R}$, and then compute the attack objective $\mathcal{O}_{atk}(\cdot)$ in

Eq. (6) (line 4-5). Second, we relax the discrete fake user behaviors $\mathbf{R}^f$ into a continuous one $\tilde{\mathbf{R}}^f \in \mathbb{R}^{|U^f| \times |I|}$, and take it as our attacker's parameter $\varphi$ (line 6). Then, we compute the gradients $\nabla_{\tilde{\mathbf{R}}^f} \mathcal{O}_{atk}(\cdot)$ according to [41] (line 7). Finally, we update the $\tilde{\mathbf{R}}^f$ and utilize a projector to map it into discrete behaviors $\mathbf{R}^f$, where each fake user can select $\tau$ items at most (line 8). Formally, given the continuous user behaviors $\tilde{\mathbf{R}}^f$, the maximum number $\tau$ of items that each fake user $u \in U^f$ can interact with, and the pre-defined threshold $th$, the projector is formulated as follows:

$$Proj(\tilde{\mathbf{R}}^f; \tau, th)[u][i] = \begin{cases} 0, & \tilde{\mathbf{R}}^f[u][i] < th \vee i \notin P_u \\ 1, & \tilde{\mathbf{R}}^f[u][i] > th \wedge i \in P_u \end{cases}, \tag{7}$$

where $P_u$ is top-$\tau$ largest item set of user $u$ according to the value in $\tilde{\mathbf{R}}^f[u]$. After finishing $L$ epochs, we output $\mathbf{R}^f$ for injection. We analyze the time complexity of Alg. 1 in Appx. A.4.

## 4 Experiments

### 4.1 Experiment Setting

**Datasets.** We evaluate our attacker on widely used three real-world datasets: MovieLens-100K (ML-100K) [14], MovieLens-1M (ML-1M) [14], and Gowalla [5]. We put the descriptions of three datasets in Appx. B.1.

**Attacker Baselines.** To verify the effectiveness of our proposed attacker, we compare our proposed attacker DADA with the three types of attackers in Tab. 1. Specifically, *heuristic-based* attackers include Random [12, 51] and Popular [29, 51], *neural network-based* attackers include TrialAttack [45], and *gradient-based* attackers include CoVis [49], PGA [20], SGLD [20], SRWA [12], TNA [11], RevAdv [41], and PAPU [51]. Particularly, we do not compare PoisonRec [38] and CopyAttack [10] in Tab. 1, because they need to access the predictions of the target RecSys for real users. Also, we compare two variants: DADA-DICT and DADA-DIV, which only optimize the difficulty and diversity-aware objective in Eq. (4) and Eq. (5), respectively.

**Targeted Recommender Systems.** Following [41, 51], we choose three representative target Rec-Sys to evaluate the attacking performance of RecSys attackers, i.e., matrix factorization-based WRMF [18], neural network-based NCF [16], and graph neural network-based LightGCN [15].

**Target Items and Evaluation Metric.** Following [41, 51], we uniformly sample 5 items as a target item set $I_t$ and measure the hit ratio Hit@N of $I_t$ on the real users $U^r$. If one of these target items appears in the top-N recommended item list of a user $u \in U^r$, the target item set is considered as a hit. Besides, we use the Normalized Discounted Cumulative Gain (NDCG) to measure the rank quality of the target items in the recommendation list. Specifically, we set the relevance score of target items and the other items for all real users as 1 and 0, respectively. Then, a higher NDCG value indicates better attack performance, where target items are ranked higher in the recommendation list. The reported results are averaged over five different runs.

**Hyperparamter Setting.** Due to datasets with different sizes, we set the fake user budget $\delta = \beta_1 \cdot |U^r|$ and the maximum number of selected items $\tau = \beta_2 \cdot |I|$, where $\beta_1$ and $\beta_2$ are the parameters to control $\delta$ and $\tau$. For default, we set $\beta_1 = 0.01$ for all datasets. For unnoticeable, we set the default $\tau$ to be similar to the average number of interacted items of real users as shown in Tab. 4, i.e., $\beta_2$ is $\{0.05, 0.05, 0.002\}$ for ML-100K, ML-1M, and Gowalla, respectively. The details about hyper-parameter settings are listed in Appx. B.2.

### 4.2 Experiment Results

We report the HR performance of attackers on ML-100K and ML-1M in Tab. 2 and Tab. 3. To better show the improvement, we add the improvement ratio of HR@10 for each attacker compared with the result in clean data. Due to space limit, we report similar results of Gowalla in Appx. B.3. We report the NDCG results on three datasets in Appx. B.4. Under the same target RecSys, the bold and underlined numbers indicate the first and second best performance, respectively.

**Baseline Comparison.** As shown in Tab. 2 and Tab. 3, our attacker DADA outperforms all baselines under different target RecSys on both ML-100K and ML-1M datasets. Specifically, heuristics-based

Table 2: Evaluation on ML-100K dataset under fake user budget $\delta = 0.01|U^r|$.

| Attacker | WRMF [18] | | | NCF [16] | | | LightGCN [15] | | |
|---|---|---|---|---|---|---|---|---|---|
| | HR@10 | HR@50 | HR@100 | HR@10 | HR@50 | HR@100 | HR@10 | HR@50 | HR@100 |
| Clean | 0.0901 | 0.3606 | 0.5677 | 0.1094 | 0.4799 | 0.7462 | 0.0977 | 0.3813 | 0.7941 |
| Random [51][12] | 0.0907 (+0.67%) | 0.3701 | 0.5536 | 0.1134 (+3.66%) | 0.5031 | 0.7511 | 0.0987 (+1.02%) | 0.3887 | 0.7824 |
| Popular [29] | 0.0912 (+1.22%) | 0.3743 | 0.5695 | 0.1145 (+4.66%) | 0.5221 | 0.7574 | 0.0998 (+2.15%) | 0.3918 | 0.7973 |
| CoVis [49] | 0.0965 (+7.10%) | 0.3722 | 0.5778 | 0.1139 (+4.11%) | 0.5585 | 0.7798 | 0.0985 (+0.82%) | 0.3960 | 0.7946 |
| TrialAttack [45] | 0.1002 (+11.21%) | 0.3902 | 0.5911 | 0.1203 (+9.96%) | 0.5998 | 0.7868 | 0.0998 (+2.15%) | 0.3992 | 0.7983 |
| SRWA [12] | 0.0906 (+0.55%) | 0.3721 | 0.5732 | 0.1172 (+7.13%) | 0.5124 | 0.7521 | 0.0991 (+1.43%) | 0.3927 | 0.7981 |
| PGA [20] | 0.0923 (+2.44%) | 0.3752 | 0.5769 | 0.1211 (+10.69%) | 0.5483 | 0.7668 | 0.0987 (+1.02%) | 0.3971 | 0.7946 |
| SGLD [20] | 0.0923 (+2.44%) | 0.3721 | 0.5716 | 0.1187 (+8.50%) | 0.5530 | 0.7675 | 0.0966 (-1.13%) | 0.3824 | 0.7952 |
| TNA [11] | 0.0974 (+8.10%) | 0.3801 | 0.5843 | 0.1199 (+9.60%) | 0.5812 | 0.7826 | 0.0924 (-5.42%) | 0.3928 | 0.7962 |
| RevAdv [41] | 0.1017 (+12.87%) | 0.4167 | 0.6285 | 0.1229 (+12.34%) | 0.6128 | 0.7885 | 0.1008 (+3.17%) | 0.4181 | 0.8026 |
| PAPU [51] | 0.1039 (+15.32%) | 0.4231 | 0.6361 | 0.1405 (+28.43%) | 0.6334 | 0.7983 | 0.1012 (+3.58%) | 0.4191 | 0.7998 |
| DADA-DICT | 0.1145 (+27.08%) | 0.4380 | 0.6471 | 0.1418 (+29.62%) | 0.6387 | 0.8130 | 0.1019 (+4.30%) | 0.4212 | 0.8061 |
| DADA-DIV | 0.1078 (+19.64%) | 0.4218 | 0.6411 | 0.1375 (+25.69%) | 0.6292 | 0.8017 | 0.1011 (+3.48%) | 0.4196 | 0.8033 |
| DADA | **0.1220 (+35.41%)** | **0.4453** | **0.6585** | **0.1513 (+38.30%)** | **0.6623** | **0.8225** | **0.1040 (+6.45%)** | **0.4286** | **0.8127** |

Table 3: Evaluation on ML-1M dataset under fake user budget $\delta = 0.01|U^r|$.

| Attacker | WRMF [18] | | | NCF [16] | | | LightGCN [15] | | |
|---|---|---|---|---|---|---|---|---|---|
| | HR@10 | HR@50 | HR@100 | HR@10 | HR@50 | HR@100 | HR@10 | HR@50 | HR@100 |
| Clean | 0.0195 | 0.0970 | 0.1912 | 0.0158 | 0.0925 | 0.1965 | 0.0257 | 0.1008 | 0.1572 |
| Random [51][12] | 0.0196 (+0.51%) | 0.1002 | 0.2030 | 0.0174 (+10.13%) | 0.0948 | 0.1983 | 0.0264 (+2.72%) | 0.1028 | 0.1618 |
| Popular [29] | 0.0196 (+0.51%) | 0.1007 | 0.2028 | 0.0202 (+27.85%) | 0.0976 | 0.2008 | 0.0268 (+4.28%) | 0.1035 | 0.1692 |
| CoVis [49] | 0.0212 (+8.72%) | 0.1061 | 0.2148 | 0.0198 (+25.32%) | 0.1073 | 0.2180 | 0.0279 (+8.56%) | 0.1057 | 0.1793 |
| TrialAttack [45] | 0.0218 (+11.79%) | 0.1084 | 0.2203 | 0.0213 (+34.81%) | 0.1088 | 0.2159 | 0.0295 (+14.79%) | 0.1115 | 0.1864 |
| SRWA [12] | 0.0202 (+3.59%) | 0.1017 | 0.2060 | 0.0194 (+22.78%) | 0.1021 | 0.2061 | 0.0272 (+5.84%) | 0.1031 | 0.1711 |
| PGA [20] | 0.0206 (+5.64%) | 0.1021 | 0.2072 | 0.0201 (+27.21%) | 0.1017 | 0.2048 | 0.0276 (+7.39%) | 0.1046 | 0.1715 |
| SGLD [20] | 0.0209 (+7.18%) | 0.1020 | 0.2025 | 0.0192 (+21.52%) | 0.0992 | 0.2008 | 0.0267 (+3.89%) | 0.1029 | 0.1702 |
| TNA [11] | 0.0221 (+13.33%) | 0.1068 | 0.2182 | 0.0195 (+23.42%) | 0.1049 | 0.2065 | 0.0282 (+9.73%) | 0.1074 | 0.1854 |
| RevAdv [41] | 0.0227 (+16.41%) | 0.1168 | 0.2412 | 0.0224 (+41.77%) | 0.1109 | 0.2197 | 0.0318 (+23.74%) | 0.1206 | 0.2037 |
| PAPU [51] | 0.0246 (+26.15%) | 0.1126 | 0.2462 | 0.0221 (+39.87%) | 0.1125 | 0.2173 | 0.0334 (+29.96%) | 0.1257 | 0.2077 |
| DADA-DICT | 0.0253 (+29.74%) | 0.1217 | 0.2581 | 0.0230 (+45.57%) | 0.1290 | 0.2219 | 0.0343 (+33.46%) | 0.1312 | 0.2184 |
| DADA-DIV | 0.0236 (+21.03%) | 0.1175 | 0.2460 | 0.0227 (+43.67%) | 0.1146 | 0.2201 | 0.0331 (+28.79%) | 0.1214 | 0.2049 |
| DADA | **0.0286 (+46.67%)** | **0.1407** | **0.2746** | **0.0233 (+47.47%)** | **0.1315** | **0.2313** | **0.0362 (+40.86%)** | **0.1407** | **0.2308** |

approaches, such as Random, Popular, and CoVis, achieve unsatisfied results. Because they only consider to increase the co-occurrence between several items and the target items, which does not relate to the attack objective directly and cannot cover various behavior patterns [41, 51, 52]. Instead, neural network-based and gradient-based attackers optimize the attack objective directly. Compared with TrialAttack, RevAdv and PAPU, several gradient-based attackers, including SRWA, PGA, SGLD, and TNA, achieve inferior performance. It is because when computing the gradients to design fake user behaviors, these approaches approximate the gradients in a biased way instead of exact computing gradients [41]. Our attacker DADA outperforms the above baselines significantly, because it enables the fake user behaviors to affect real users in a diverse manner.

**Ablation Study.** As shown in Tab. 2 and Tab. 3, DADA-DICT achieves the second best results, outperforming all baselines. It is because the difficulty-aware attack objective allows DADA-DICT to concentrate on easy users with high tendencies toward the target items. Moreover, DADA-DIV achieves satisfied performance. It outperforms current state-of-the-art attackers, including RevAdv and PAPU, in most cases. It is because the diversity-aware attack objective allows the generated fake user behaviors to affect real users more diversely, avoiding large communities dominating the generated fake users. Nevertheless, DADA-DIV does not outperform DADA-DICT, because DADA-DIV cannot differentiate easy users from all real users and put more efforts on them. Furthermore, DADA that considers both user difficulty and diversity can perform the best. In such a way, it can potentially affect easy users from different communities, thereby boosting attacker's effectiveness.

### 4.3 Parameter Sensitivity

In this subsection, we evaluate DADA under different parameters. As shown in Tab 2 and Tab. 3, the results of DADA and baselines on different target RecSys and hit ratios have similar trends. For clarification, we use WRMF and HR@50 for evaluation on ML-100K and ML-1M datasets.

**Fake User Budget $\delta$.** We study the effect of fake user budget $\delta = \beta_1|U^r|$ by varying $\beta_1 \in \{0.01, 0.03, 0.05, 0.07, 0.09\}$. For clarification, we only compare the effective baselines, including TrialAttack, TNA, RevAdv, and PAPU. As shown in Fig. 2 (a) and (b), HR@50 of all attackers increases as the fake user budget increases on both ML-100K and ML-1M datasets. Furthermore,

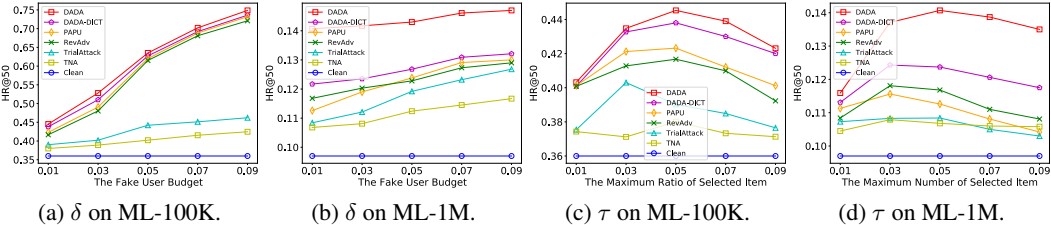

|  (a) $\delta$ on ML-100K. | (b) $\delta$ on ML-1M. | (c) $\tau$ on ML-100K. | (d) $\tau$ on ML-1M. |

Figure 2: Evaluation on the fake user budget and maximum selected items.

DADA outperforms all baselines under different attack budgets, i.e., the red solid line is above other lines, demonstrating its effectiveness. Particularly, as the fake user budget increases, our attacker and baselines improve HR@50 on ML-100K more than that on ML-1M. The reason is that ML-100K only has 1682 items, while ML-1M has 3706 items. As a result, it is more difficult to promote target items into the top-50 recommendation list of each user in ML-1M than ML-100K.

**Maximum Selected Items $\tau$.** We study the effect of different maximum number of selected items $\tau = \beta_2 |I|$ by varying $\beta_2 \in \{0.01, 0.03, 0.05, 0.07, 0.09\}$. As shown in Fig. 2 (c) and (d), as the maximum number of items increases, the performance of attackers first increases and then decreases on both datasets. It indicates that selecting suitable items can increase the attacker's power. But more items will reduce the effectiveness of attackers, since the valuable items are limited, and extra items will introduce noise [12, 29, 51]. Furthermore, compared with these powerful baselines, DADA can achieve stable and the best performance under different maximum number of selected items.

### 4.4 More Insights via Empirical Study

Due to space limit, we include more experiments in Appx. B for more insights. In Appx. B.5, to better mimic the real-world applications, we demonstrate the generality and practicability of our attacker on partial and noisy user behaviors. In Appx. B.6, we evaluate our attacker under different item popularity. In Appx. B.7, we more comprehensively reveal the effectiveness of the difficulty-aware objective and quantitatively measure the diversity brought by the diversity-aware objective. In Appx. B.8, we evaluate our attacker under more parameter settings, including extremely limited fake user budget and $\lambda$ in Eq (6). In Appx. B.9, we analyze the hit ratio improvement under different target RecSys. In Appx. B.10, we provide visualizations on gradients provided by different users and fake user distributions. Finally, in Appx. B.11, we list several future directions based on our observations.

## 5 Conclusion

In this paper, we revisit the problem of injective attacks against recommender systems and reveal two issues of existing attackers: difficulty-agnostic and diversity-deficit issues. To address these two issues, we propose DADA, a difficulty and diversity aware attacker. Within a limited fake user budget, DADA enables the generated fake user behaviors to concentrate on easy users who have high tendencies toward the target item, and affect real users in a more diverse manner. Extensive experiments on three real-world datasets demonstrate the effectiveness of our proposed attacker. In this paper, we mainly reveal the weakness of existing recommender systems and discuss how to attack them. For the sake of society, we discuss how to defense and resist such attacks in Appx. B.12.

## 6 Acknowledgement

Lei Chen's work is partially supported by National Key Research and Development Program of China Grant No. 2022YFE0200500 and 2018AAA0101100, the Hong Kong RGC GRF Project 16213620, CRF Project C6030-18G, C1031-18G, C5026-18G, CRF C2004-21GF, AOE Project AoE/E603/18, RIF Project R6020-19, Theme-based project TRS T41-603/20R, China NSFC No. 61729201, Guangdong Basic and Applied Basic Research Foundation 2019B151530001, Hong Kong ITC ITF grants ITS/044/18FX and ITS/470/18FX, Microsoft Research Asia Collaborative Research Grant, HKUST-NAVER/LINE AI Lab, Didi-HKUST joint research lab, HKUST-Webank joint research lab grants and HKUST Global Strategic Partnership Fund (2021 SJTU-HKUST). Shimin Di's work is supported by the JC STEM Lab of Data Science Foundations funded by The Hong Kong Jockey Club Charities Trust.

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
