# A Appendix

## A.1 Additional Motivational Observations

### A.1.1 Additional Results on Difficulty-agnostic Analysis

Fig. 3 shows additional results of TNA [11] and PAPU [51] on ML-100K [14]. We follow the same experiment setting in Sec 3.1.1, and show the average loss $\frac{\sum_{u \in U^g} \mathcal{L}_{atk}(u,t)}{|U^g|}$, the average gradient $\frac{\sum_{u \in U^g} \|\nabla_\varphi \mathcal{L}_{atk}(u,t)\|_1}{|U^g| \cdot (|U^f| \cdot |I|)}$, and the HR@50 on each user group. As shown in Fig. 3, TNA and PAPU put more efforts (i.e. greater loss and gradients) into more difficult users, while only HR@50 of the easy group increases and the medium and difficult groups perform poorly. The results also verify the conclusion in the Sec 3.1.1: it is simpler to promote the target item to easier users than more difficult ones, and putting more efforts on more difficult users will reduce the effectiveness of attackers. Also, we do not show the result of TrialAttack [45], since it satisfies the *difficulty-property 1* that enables attackers to put more efforts on easy users (see Sec. 3.1.2).

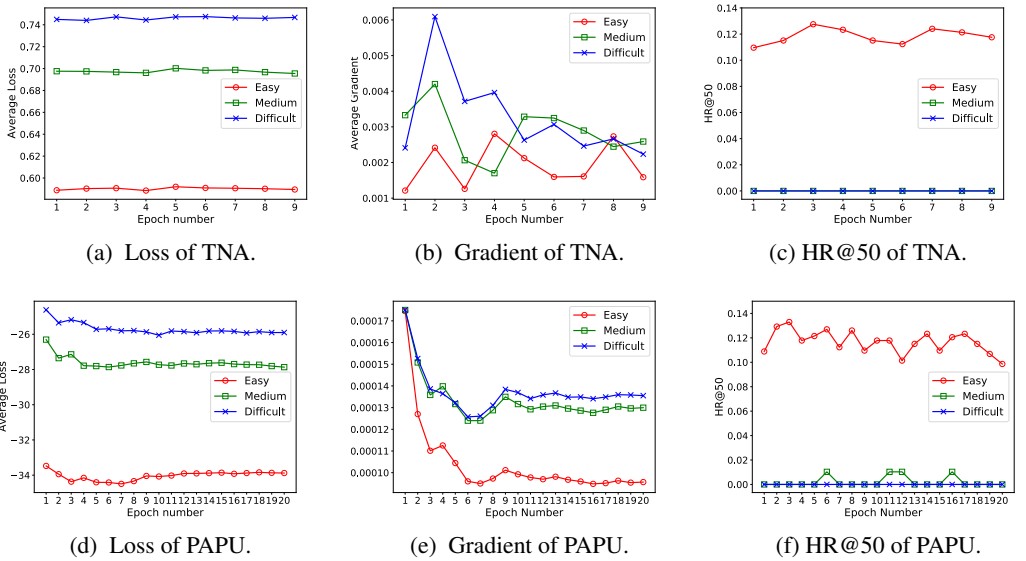

(a) Loss of TNA.     (b) Gradient of TNA.     (c) HR@50 of TNA.

(d) Loss of PAPU.     (e) Gradient of PAPU.     (f) HR@50 of PAPU.

Figure 3: Additional results for difficulty-agnostic analysis.

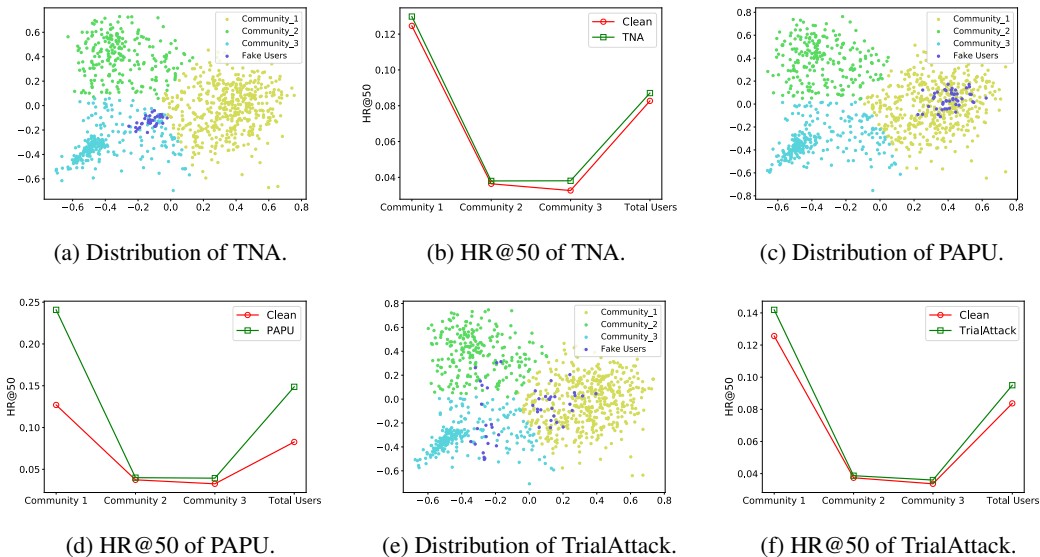

(a) Distribution of TNA.     (b) HR@50 of TNA.     (c) Distribution of PAPU.

(d) HR@50 of PAPU.     (e) Distribution of TrialAttack.     (f) HR@50 of TrialAttack.

Figure 4: Additional results for diversity-deficit analysis.

### A.1.2 Additional Results on Diversity-agnostic Analysis

Fig. 4 shows additional results of TNA [11], PAPU [51], and TrialAttack [45] on ML-100K [14]. We follow the same experiment setting in Sec 3.2.1. As shown in Fig. 4 (a) and (c), the fake users of TNA and PAPU form a cluster that is distributed in community 3 and community 1, respectively. Consequently, in Fig. 4 (b) and (d), TNA and PAPU improve HR@50 on community 3 and 1, respectively, while keeping similar HR@50 on the other communities. Thus, they suffer from the diversity-deficit issue. Particularly, in Fig. 4 (e), the fake users of TrialAttack are dispersed diversely among real users. However, in Fig. 4 (f), TrialAttack only achieves improvements in community 1, while achieving similar performance in the other communities. It is because TrialAttack uses diverse noise from various communities as the inputs of generative neural networks, forcing the generated fake users to disperse diversely. Nevertheless, since its attack objective cannot guarantee fake users to affect real users diversely, it still cannot improve the performance for each community. Thus, TrialAttack meets the diversity-deficit issue as well.

## A.2 Proof

### A.2.1 Proof of Proposition 1

*Proof.* The derivative of Eq. (4) is as follows:

$$\frac{\partial Dict(\hat{\mathcal{M}}_{\hat{\theta}^*}(\mathbf{R}); u, t)}{\partial \hat{\mathbf{S}}[u][t]} = \begin{cases} \dfrac{b_1}{(1 - b_1 \cdot \hat{\mathbf{S}}[u][t]) \cdot \ln 2}, & t \notin \Gamma_u \\ 0, & t \in \Gamma_u \end{cases},$$

Given users $u$ and $v$, and $b_1 > 0$, if $\hat{\mathbf{S}}[v][t] < \hat{\mathbf{S}}[u][t]$ and $t \notin \Gamma_u$, $\frac{\partial Dict(v,t)}{\partial \hat{\mathbf{S}}[v][t]} < \frac{\partial Dict(u,t)}{\partial \hat{\mathbf{S}}[u][t]}$. Also, when $t \in \Gamma_u$, the derivative is 0. Thus, our difficulty-aware attack objective satisfy both properties. $\square$

### A.2.2 Proof of Proposition 2

*Proof.* The derivative of Eq. (5) is as follows:

$$\frac{\partial Div(\hat{\mathbf{S}}[u][t]; u, t, \hat{\mathbf{S}}'[u][t])}{\partial \triangle \hat{\mathbf{S}}[u][t]} = \begin{cases} \dfrac{b_2 \exp(-b_2(\triangle \hat{\mathbf{S}}[u][t] - c))}{(1 + \exp(-b_2(\triangle \hat{\mathbf{S}}[u][t] - c)))^2} & t \notin \Gamma_u \\ 0, & t \in \Gamma_u \end{cases},$$

Given users $u$ and $v$, and $b_2 > 0$, if $\triangle \hat{\mathbf{S}}[v][t] > \triangle \hat{\mathbf{S}}[u][t]$ and $t \notin \Gamma_u$, $\frac{\partial Div(v,t)}{\partial \triangle \hat{\mathbf{S}}[v][t]} < \frac{\partial Div(u,t)}{\partial \triangle \hat{\mathbf{S}}[u][t]}$. Also, when $t \in \Gamma_u$, the derivative is 0. Thus, our diversity-aware attack objective satisfy both properties. $\square$

## A.3 Gradient-based Algorithm

---
**Algorithm 1:** Fake User Behaviors Generation

---
**Input:** Real user behaviors $\mathbf{R}^r$, the surrogate RecSys $\hat{\mathcal{M}}_{\hat{\theta}}$, the target item $t$, fake user budget $|U^f|$, the maximum number of interacted item $\tau$, the threshold $th$, the epoch number $L$, and learning rate $\alpha$.

**Output:** Fake use behaviors $\mathbf{R}^f \in \{0, 1\}^{|U^f| \times |I|}$

1   Initialize $\mathbf{R}^f \in \{0, 1\}^{|U^f| \times |I|}$
2   **for** $l = 1$ **to** $L$ **do**
3      $\mathbf{R} = [\mathbf{R}^r, \mathbf{R}^f]$
4      $\hat{\theta}^* = \arg\min_{\hat{\theta}} \mathcal{L}_{tra}(\hat{\mathcal{M}}_{\hat{\theta}}(\mathbf{R}), \mathbf{R})$
5      Compute $\mathcal{O}_{atk}(\cdot) \leftarrow$ Eq. (6)
6      $\tilde{\mathbf{R}}^f = \textbf{Continuous}(\mathbf{R}^f)$
7      $\tilde{\mathbf{R}}^f = \tilde{\mathbf{R}}^f + \alpha \cdot \nabla_{\tilde{\mathbf{R}}^f} \mathcal{O}_{atk}(\cdot)$
8      $\mathbf{R}^f = Proj(\tilde{\mathbf{R}}^f; \tau, th)$
9   **Return** $\mathbf{R}^f$

---

Table 4: The statistics of three datasets.

|          | $|U^r|$ | $|I|$ | **Avg** $|I_u|$ | $\|\mathbf{R}^r\|_0$ | **Sparsity** |
| -------- | ------- | ----- | --------------- | -------------------- | ------------ |
| **ML-100K** | 943  | 1682  | 6.30%           | 100,000              | 93.70%       |
| **ML-1M**   | 6040 | 3706  | 4.47%           | 1000,209             | 95.53%       |
| **Gowalla** | 13149| 14007 | 0.24%           | 433,356              | 99.76%       |

Table 5: The configuration of the surrogate and target RecSys.

| RecSys | Configuration |
| ------ | ------------- |
| Surrogate-WRMF | Latent dimension: 128; Learning rate: 1e-2; Weights for positive feedback: 20; $L_2$ regularization coefficient: 1e-5; Training epochs: 100; |
| WRMF | Latent dimension: 128; Learning rate: 1e-2; Weights for positive feedback: 20; $L_2$ regularization coefficient: 1e-5; Training epochs: 100; |
| NCF | Latent dimension: 128; Learning rate: 1e-4; $L_2$ regularization coefficient: 1e-5; Training epochs: 200; |
| LightGCN | Latent dimension: 128; Learning rate: 1e-4; $L_2$ regularization coefficient: 1e-5; Layer number: 2; Training epochs: 200; |

### A.4 Greedy Algorithm Alg. 1.

We propose Alg. 1 to optimize our proposed attack objective in Eq. (6). We analyze its time complexity as follows. Assuming that we train the surrogate RecSys $\hat{\mathcal{M}}_{\hat{\theta}}(\mathbf{R})$ $m$ times in each attack epoch $l$. In each attack epoch, it takes $O(m|\hat{\theta}||\mathbf{R}|)$ time for training the surrogate RecSys (line 4). Moreover, we compute $\nabla_{\tilde{\mathbf{R}}^f}\mathcal{O}_{atk} = \nabla_{\hat{\theta}^m}\mathcal{O}_{atk} \cdot \nabla_{\hat{\theta}^{m-1}}\hat{\theta}^m \cdots \nabla_{\tilde{\mathbf{R}}^f}\hat{\theta}^1$ (line 7). Based on the reverse-model algorithm differentiation [3], we need to accumulate the gradients of $\hat{\theta}$ for $m$ times to compute $\nabla_{\tilde{\mathbf{R}}^f}\mathcal{O}_{atk}$, which takes $O(m|\hat{\theta}|)$ time [41, 52]. Therefore, the total time complexity is $O(Lm|\hat{\theta}||\mathbf{R}|)$.

## B Experiment Supplementary

### B.1 Dataset Description

Following [12, 41, 44, 45, 51], we use three popular datasets for evaluation. The statistics are shown in Tab. 4.

- **MovieLens-100K (ML-100K)** and **MovieLens-1M (ML-1M)** [14]: They are widely used movie recommendation datasets. Following [31, 51], we transform numerical ratings into implicit feedback (1 for positive interaction, 0 for negative interaction).
- **Gowalla** [5]: It consists of implicit check-in feedbacks between users and venues. Following [40, 41], we remove cold-start users and items that have less than 15 feedbacks.

### B.2 Additional Experiment and Hyperparameter Setting

**Attack Objective Setting.** In the main experiments, we select five target items for evaluation, and the attack objective is the sum of the attack objectives on each target item $t \in I_t$ in Eq. (2). Formally, given multiple target items $I_t$, the combined attack objective on multiple target items $I_t$ can be computed as follows:

$$\max_{\mathbf{R}^f} \sum_{u \in U^r} \sum_{t \in I_t} Dict(\hat{\mathcal{M}}_{\theta^*}(\mathbf{R}); u, t) + \lambda \cdot Div(\hat{\mathbf{S}}[u][t]; u, t, \hat{\mathbf{S}}'[u][t]),$$

Then, if one target item $t$ is in the recommendation list of user $u$, we set the attack objective value of $u$ regarding $t$ as 0. In other words, we only zero out the gradients of a user $u$ until all target items are in its top-N recommendation list.

**Hyperparameter Setting.** For attacker baselines, we use their default hyperparameter settings. For our attacker, we search the hyperparameter $b_1$ in Eq. (4) and $b_2$ in Eq. (5) from $\{100, 200, 300, 400, 500\}$ and search $\lambda$ in Eq. (6) from $\{10, 20, 30, 40, 50\}$ for three datasets. We set

Table 6: Evaluation on Gowalla dataset under fake user budget $\delta = 0.01|U^r|$.

| Attacker | WRMF [18] | | | NCF [16] | | | LightGCN [15] | | |
|---|---|---|---|---|---|---|---|---|---|
| | HR@10 | HR@50 | HR@100 | HR@10 | HR@50 | HR@100 | HR@10 | HR@50 | HR@100 |
| Clean | 0.0007 | 0.0052 | 0.0142 | 0.0019 | 0.0062 | 0.0267 | 0.0013 | 0.0056 | 0.0181 |
| Random [51, 12] | 0.0010 (+42.86%) | 0.0071 | 0.0192 | 0.0025 (+31.58%) | 0.0097 | 0.0277 | 0.0017 (+30.77%) | 0.0092 | 0.0226 |
| Popular [29] | 0.0013 (+85.71%) | 0.0098 | 0.0207 | 0.0031 (+63.16%) | 0.0095 | 0.0283 | 0.0021 (+61.54%) | 0.0112 | 0.0243 |
| CoVis [49] | 0.0016 (+128.57%) | 0.0112 | 0.0217 | 0.0037 (+94.74%) | 0.0116 | 0.0358 | 0.0022 (+69.23%) | 0.0129 | 0.0282 |
| TrialAttack [45] | 0.0010 (+42.86%) | 0.0105 | 0.0221 | 0.0042 (+121.05%) | 0.0106 | 0.0351 | 0.0022 (+69.23%) | 0.0117 | 0.0294 |
| SRWA [12] | 0.0012 (+71.43%) | 0.0086 | 0.0175 | 0.0037 (+94.74%) | 0.0092 | 0.0347 | 0.0017 (+30.77%) | 0.0106 | 0.0241 |
| PGA [20] | 0.0012 (+71.43%) | 0.0084 | 0.0183 | 0.0029 (+52.63%) | 0.0087 | 0.0319 | 0.0017 (+30.77%) | 0.0092 | 0.0229 |
| SGLD [20] | 0.0008 (+14.29%) | 0.0078 | 0.0172 | 0.0031 (+63.16%) | 0.0073 | 0.0327 | 0.0021 (+61.54%) | 0.0087 | 0.0223 |
| TNA [11] | 0.0011 (+57.14%) | 0.0109 | 0.0215 | 0.0036 (+89.47%) | 0.0102 | 0.0328 | 0.0009 (-30.77%) | 0.0106 | 0.0268 |
| RevAdv [41] | 0.0017 +(142.86%) | 0.0121 | 0.0281 | 0.0032 (+68.42%) | 0.0143 | 0.0355 | 0.0019 (+46.15%) | 0.0138 | 0.0364 |
| PAPU [51] | 0.0021 (+200.00%) | 0.0125 | 0.0282 | 0.0043 (+126.32%) | 0.0161 | 0.0395 | 0.0025 (+92.31%) | 0.0165 | 0.0373 |
| DADA-DICT | 0.0027 (+285.71%) | 0.0134 | 0.0296 | 0.0051 (+168.42%) | 0.0168 | 0.0442 | 0.0031 (+138.46%) | 0.0175 | 0.0407 |
| DADA-DIV | 0.0020 (+185.71%) | 0.0129 | 0.0288 | 0.0039 (+105.26%) | 0.0156 | 0.0382 | 0.0023 (+76.92%) | 0.0142 | 0.0360 |
| DADA | **0.0033 (+371.43%)** | **0.0157** | **0.0316** | **0.0064 (+236.84%)** | **0.0198** | **0.0486** | **0.0040 (+207.69%)** | **0.0217** | **0.0451** |

Table 7: Evaluation on NDCG@50 under fake user budget $\delta = 0.01|U^r|$.

| Attacker | ML100K | | | ML-1M | | | Gowalla | | |
|---|---|---|---|---|---|---|---|---|---|
| | WRMF | NCF | LightGCN | WRMF | NCF | LightGCN | WRMF | NCF | LightGCN |
| Clean | 0.0450 | 0.0312 | 0.0360 | 0.0089 | 0.0082 | 0.0094 | 0.0004 | 0.0004 | 0.0006 |
| Random [51, 12] | 0.0447 (-0.67%) | 0.0552 | 0.0370 | 0.0099 (+11.24%) | 0.0084 | 0.0097 | 0.0005 (+25.00%) | 0.0006 | 0.0007 |
| Popular [29] | 0.0453 (+0.67%) | 0.0571 | 0.0371 | 0.0100 (+12.36%) | 0.0087 | 0.0096 | 0.0006 (+50.00%) | 0.0006 | 0.0007 |
| CoVis [49] | 0.0458 (+1.78%) | 0.0588 | 0.0389 | 0.0101 (+13.48%) | 0.0094 | 0.0103 | 0.0008 (+100.00%) | 0.0009 | 0.0010 |
| TrialAttack [45] | 0.0478 (+6.22%) | 0.0601 | 0.0403 | 0.0105 (+17.98%) | 0.0093 | 0.0105 | 0.0011 (+175.00%) | 0.0007 | 0.0010 |
| SRWA [12] | 0.0471 (+4.67%) | 0.0565 | 0.0390 | 0.0094 (+5.62%) | 0.0088 | 0.0107 | 0.0006 (+50.00%) | 0.0007 | 0.0008 |
| PGA [20] | 0.0464 (+3.11%) | 0.0558 | 0.0374 | 0.0097 (+8.99%) | 0.0092 | 0.0097 | 0.0006 (+50.00%) | 0.0007 | 0.0008 |
| SGLD [20] | 0.0469 (+4.22%) | 0.0589 | 0.0371 | 0.0095 (+6.74%) | 0.0089 | 0.0101 | 0.0006 (+50.00%) | 0.0009 | 0.0009 |
| TNA [11] | 0.0476 (+5.78%) | 0.0605 | 0.0389 | 0.0103 (+15.73%) | 0.0090 | 0.0107 | 0.0009 (+125.00%) | 0.0008 | 0.0010 |
| RevAdv [41] | 0.0560 (+24.44%) | 0.0610 | 0.0415 | 0.0107 (+20.22%) | 0.0109 | 0.0124 | 0.0016 (+300.00%) | 0.0014 | 0.0013 |
| PAPU [51] | 0.0566 (+25.78%) | 0.0623 | 0.0404 | 0.0109 (+22.47%) | 0.0116 | 0.0136 | 0.0018 (+350.00%) | 0.0015 | 0.0015 |
| DADA-DICT | 0.0581 (+29.11%) | 0.0651 | 0.0433 | 0.0131 (+47.19%) | 0.0129 | 0.0139 | 0.0028 (+600.00%) | 0.0017 | 0.0016 |
| DADA-DIV | 0.0541 (+20.22%) | 0.0625 | 0.0415 | 0.0114 (+28.09%) | 0.0121 | 0.0129 | 0.0023 (+475.00%) | 0.0015 | 0.0014 |
| DADA | **0.0612 (+36.00%)** | **0.0697** | **0.0459** | **0.0136 (+52.81%)** | **0.0142** | **0.0146** | **0.0032 (+700.00%)** | **0.0021** | **0.0018** |

the learning rate $\alpha$ in Alg. 1 as 1, and set the threshold $th$ in Eq. (7) as 0.05. Following [11, 20, 41, 51], we select the representative WRMF [18] as the surrogate RecSys. The default parameters of the surrogate and target RecSys are listed in Tab. 5.

Also, all experiments are conducted on CentOS 7 machines with a 20-core Intel(R) Xeon(R) Silver 4210 CPU @ 2.20GHz, 8 NVIDIA GeForce RTX 2080 Ti GPUs (11G), and 92G of RAM.

## B.3 Additional Results on Gowalla

In this subsection, we show additional results of attackers on Gowalla dataset in Tab. 6. Specifically, similar to Tab. 2 and Tab. 3, our attacker DADA and the variant DADA-DICT outperform all the baselines, demonstrating their effectiveness. Also, the hit ratio of the target item on the Gowalla is lower than that on the ML-100K and ML-1M, owing to the fact that Gowalla's item number (14007) is significantly higher than the other two datasets.

## B.4 Evaluation on NDCG Metric

In this subsection, we use the NDCG metric to evaluate the rank quality of target items in the top-N recommendation list of users. We report the results of NDCG@50 of our attacker and baselines on three datasets under a fake user ratio of 0.01, since results on NDCG@10 and NDCG@100 have a similar trend to NDCG@50. As shown in Tab. 7, heuristics-based attackers, such as Random, Popular, and CoVis, achieve unsatisfied results on NDCG@50. It is because such heuristic-based baselines cannot optimize the attack objective directly. Besides, compared with TrialAttack, RevAdv, and PAPU, several gradient-based attackers, including SRWA, PGA, SGLD, and TNA, achieve inferior performance on NDCG. It is because they compute the gradients in a biased way, thus damaging the rank quality of target items [41, 51]. Finally, our proposed attacker DADA achieves the best performance compared to baselines, demonstrating the higher rank quality of target items.

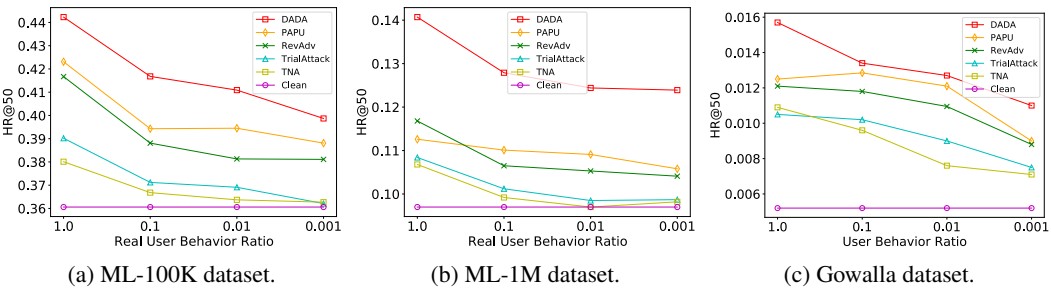

(a) ML-100K dataset.  (b) ML-1M dataset.  (c) Gowalla dataset.

Figure 5: Evaluation on partial real user behaviors.

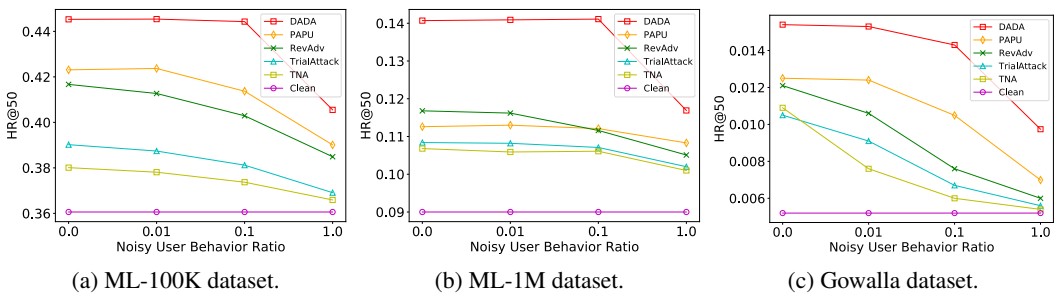

(a) ML-100K dataset.  (b) ML-1M dataset.  (c) Gowalla dataset.

Figure 6: Evaluation on noisy user behaviors.

## B.5 Partial and Noisy User Behaviors

In the real world, the real user behaviors collected by attackers from open platforms may not complete or may have noise. In this subsection, to show the general practicability of our proposed attacker DADA, we evaluate DADA under partial user behaviors and noisy user behaviors. Since the results of DADA and baselines on different target RecSys and different hit ratios have similar trends as shown in Tab 2, Tab. 3, and Tab. 6, we use WRMF as the target RecSys and use HR@50 as the evaluation metric. Also, we only compare the effective baselines, including TrialAttack, TNA, RevAdv, and PAPU. Finally, we discuss how to collect the real user behaviors from real-world platforms.

**Partial User Behaviors.** We set the size of real user behaviors for training as $||\mathbf{R}^p||_0 = \beta_3||\mathbf{R}^r||_0$, and vary the real user behavior ratio by $\beta_3 \in \{1, 0.1, 0.01, 0.001\}$. Specifically, given real user behaviors $\mathbf{R}^r$ and the ratio $\beta_3$, we randomly sample $\beta_3||\mathbf{R}^r||_0$ user-item interactions (non-zero elements) from $\mathbf{R}^r$ to construct the partial user behaviors $\mathbf{R}^p$.

As shown in Fig. 5, as the ratio of user behaviors decreases, the HR@50 of all attackers on three datasets decreases. It is because a low ratio of user behaviors reduces the surrogate RecSys's ability to mimic the target RecSys, i.e., the suboptimal surrogate RecSys cannot reflect the tendency between users and items accurately as the target RecSys. Thus, it impedes the effective fake user behaviors generation. In addition, our proposed attacker DADA outperforms all baselines under various user behavior ratios, as indicated by the red solid lines being above the other lines, demonstrating its effectiveness. It is because our attacker attempts to maximize the positive difference between the tendency after and before attacks regarding the target items for each user. Such a way treats all users equally, regardless of their tendency towards the target items learned by the surrogate RecSys that is trained on the partial user behaviors. Furthermore, DADA can achieve satisfactory performance even with only $0.001||\mathbf{R}^r||_0$ real user behaviors as the training data, demonstrating its practicability.

**Noisy User Behaviors.** Following [51], we add noise into real user behaviors by randomly flipping the interaction behavior between users and items. Specifically, we set the size of added noisy user behaviors as $\beta_4||\mathbf{R}^r||_0$, and vary the added noisy user behavior ratio by $\beta_4 \in \{0, 0.01, 0.1, 1\}$. Specifically, given user set $U^r$, item set $I$, and noise behavior ratio $\beta_4$, we randomly sample $\beta_1||\mathbf{R}^r||_0$ pairs of users and items from $U^r$ and $I$. Then, for each sampled user-item pair $(u, i)$, we set $\mathbf{R}^r[u][i] = 0$ if $\mathbf{R}^r[u][i] = 1$, or else we set $\mathbf{R}^r[u][i] = 1$ if $\mathbf{R}^r[u][i] = 0$.

As shown in Fig. 6, as the ratio of added noise user behaviors increases, the HR@50 of attackers on three datasets decreases. Similar to training on partial user behaviors, the suboptimal surrogate

Table 8: Evaluation on Target Item with Different Popularity

| Attacker | Target Item Popularity | | | | |
|---|---|---|---|---|---|
| | Most Pop. | Popular | Ordinary | Unpopular | Most Unp. |
| Clean | 0.5421 | 0.2471 | 0.0764 | 0.0025 | 0.0000 |
| Random [12, 51] | 0.5435 | 0.2513 | 0.0806 | 0.0043 | 0.0006 |
| Popular [29] | 0.5422 | 0.2524 | 0.0774 | 0.0045 | 0.0006 |
| CoVis [49] | 0.5412 | 0.2534 | 0.0838 | 0.0036 | 0.0007 |
| TrialAttack [45] | 0.5672 | 0.2624 | 0.1026 | 0.0040 | 0.0005 |
| SRWA [12] | 0.5465 | 0.2571 | 0.0827 | 0.0037 | 0.0003 |
| PGA [20] | 0.5594 | 0.2515 | 0.0812 | 0.0032 | 0.0003 |
| SGLD [20] | 0.5584 | 0.2491 | 0.0798 | 0.0030 | 0.0001 |
| TNA [11] | 0.5599 | 0.2583 | 0.0812 | 0.0041 | 0.0007 |
| RevAdv [41] | 0.5839 | 0.2880 | 0.1448 | 0.0058 | 0.0006 |
| PAPU [51] | 0.5921 | 0.2922 | 0.1491 | 0.0055 | 0.0008 |
| DADA-DICT | 0.6094 | 0.2990 | 0.1622 | 0.0086 | 0.0013 |
| DADA-DIV | 0.5846 | 0.2916 | 0.1507 | 0.0076 | 0.0010 |
| DADA | **0.6116** | **0.3033** | **0.1654** | **0.0088** | **0.0015** |

RecSys trained on noise user behaviors may not learn the accurate latent tendency between users and items accurately as the target RecSys. It impedes attackers to generate effective fake user behaviors. Particularly, under the noisy user behavior ratio of 1.0, the HR@50 of attackers decreases dramatically, but the fake user behaviors generated by attackers can still affect the target RecSys to promote the target items to more real users. It is because the added noise user behaviors are sampled from all pairs between users and items, and thus the added noise user behaviors of ratio 1.0 may not completely change the real user behaviors (non-zero elements). Also, our proposed attacker DADA outperforms all baselines under various noise user behavior ratios, as indicated by the red solid lines being above the other lines. Similar to the partial user behaviors scenario, it is because our attacker wants to maximize the positive difference between the tendency after and before attacks for each user regarding the target items. In this manner, all users are treated equally, regardless of their tendency towards the target items learned by the suboptimal surrogate RecSys. Additionally, DADA can achieve superior performance than baselines under the noisy user behavior ratio of 1.0, demonstrating its practicability.

**User Behavior Collection Discussion.** Basically, we can collect user behaviors from social media platforms (e.g., IMDB[2], rottentomatoes[3], Goodreads[4], and YouTube[5]) and e-commerce platforms (e.g., Amazon[6] and Yelp[7]). These platforms allow us to access the item interaction histories of users or the comments of users on items. Then, we can construct user behaviors by crawling the rating histories of users from their profiles and reviews on items. Particularly, even though the collected user behaviors may be incomplete and have some noise, they can be used to generate effective fake user behaviors. It is because they can mimic the real distributions of all users.

## B.6 Evaluation on Item Popularity

Note that we have evaluated target items randomly selected from all items in Sec. 4.2 and Appx. B.3. In this subsection, we comprehensively analyze the attacker on the target items with various popularity. Following [51], we define Most Popular (Most Pop.) items whose click number (#click) is above 80 percentile. Also, we can define Popular (80 percentile $\geq$ #click > 60 percentile), Ordinary (60 percentile $\geq$ #click > 40 percentile), Unpopular (40 percentile $\geq$ #click > 20 percentile), and Most Unpop. (#click $\leq$ 20 percentile). As shown in Tab 2, Tab. 3, and Tab. 6, the results of DADA and

---

[2]https://www.imdb.com/

[3]https://www.rottentomatoes.com/

[4]https://www.goodreads.com/

[5]https://www.youtube.com/

[6]https://www.amazon.com.au/

[7]https://www.yelp.com/

Table 9: Ablation study on difficulty-aware objective.

| | ML-100K | ML-1M | Gowalla |
|---|---|---|---|
| TrialAttack | 0.3902 | 0.1084 | 0.0105 |
| TrialAttack++ | 0.3976 (+1.90%) | 0.1093 (+0.83%) | 0.0107 (+1.90%) |
| TNA | 0.3801 | 0.1068 | 0.0109 |
| TNA++ | 0.3892 (+2.39%) | 0.1076 (+0.75%) | 0.0112 (+2.75%) |
| RevAdv | 0.4167 | 0.1168 | 0.0121 |
| RevAdv++ | 0.4231 (+1.54%) | 0.1210 (+3.60%) | 0.0122 (+0.83%) |
| PAPU | 0.4224 | 0.1126 | 0.0125 |
| PAPU++ | 0.4284 (+1.42%) | 0.1195 (+6.13%) | 0.0129 (+3.20%) |
| DADA-DICT | **0.4380** | **0.1217** | **0.0134** |

baselines on different target RecSys and hit ratios have similar trends. Thus, we conduct experiments on ML-1M, and report the average HR@50 on five runs against the target WRMF RecSys.

As shown in Tab. 8, as the popularity of the target items decreases, the hit ratio of them also decreases. It is because as the target items become less popular, the tendency of real users toward these target items deceases as well. Thus, it is more difficult to affect the target RecSys to recommend these unpopular items to real users. Besides, DADA and DADA-DICT outperform all baselines under different popularity, demonstrating their effectiveness. More importantly, as the popularity of the target items decreases, DADA can achieve more improvements over the best baseline PAPU. For example, Over PAPU, DADA achieves a 3.3% improvement on Most Popular items, while achieving 87.5% improvement on Most Unpopular items. It is because compared with popular target items, there are fewer real users that have a high tendency toward these unpopular target items. Thus, our attacker that focuses on easy users diversely can achieve more improvements on these unpopular target items.

## B.7 Additional Ablation Study

In this subsection, we conduct two ablation studies to further investigate our proposed difficulty-aware and diversity-aware attack objectives. Due to similar observations, we use WRMF as the target RecSys and use HR@50 as the evaluation metric. Also, for clarification, we only compare the effective baselines, including TrialAttack, TNA, RevAdv, and PAPU.

**Difficulty-aware Objective.** As shown in Eq. (4), there are two cases in our difficulty-aware objective $Dict(\cdot)$: (1) if the target item $t$ is not successfully promoted to the user $u$, i.e., $t \notin \Gamma_u$, we enable $u$ to dominate the gradient computation more than more difficult users. (2) If $t \in \Gamma_u$, we directly zero out the gradients provided by the user $u$. To further reveal the promotion gain from these two cases, we compare the baseline variants by setting the gradients of successfully promoted users as 0. The variant of each [baseline] is denoted as [baseline]++. The experiment setting is the default experiment setting in Appx. B.2.

As illustrated in Tab. 9, the variant of each baseline outperforms the original baseline on three datasets. It indicates that zeroing out the gradients of successfully promoted users whose recommendation lists contain the target item is beneficial, enabling attackers to generate fake users that affect more users. Moreover, DADA-DICT outperforms all the variants of baselines. It demonstrates that, except for zero-outing the gradients of successfully promoted users, it is preferable to concentrate on easy users, as it is easier to promote the target item for them. In summary, both cases in our difficulty-aware objective can improve the promotion gain.

**Diversity-aware Objective.** To address the *diversity-deficit* issue, we expect that the attackers create fake user behaviors that can manipulate the target RecSys to recommend the target into the top-N list of users in different communities. Based on this, we can qualitatively measure the diversity brought by each attacker based on the hit ratio improvement of each community. Formally, given communities $\{C_j\}_{j=1}^{n_c}$ and a target item $t$, we denote the hit ratio improvement of each community $C_j$ after injecting fake behaviors as $\triangle HR(C_j)$, and normalized hit ratio difference can be computed as

Table 10: Ablation study on diversity-aware objective.

| Attacker | ML-100K | | | | | ML-1M | | | | | Gowalla | | | | |
|---|---|---|---|---|---|---|---|---|---|---|---|---|---|---|---|
| | $C_1$ | $C_2$ | $C_3$ | All | $D$ | $C_1$ | $C_2$ | $C_3$ | All | $D$ | $C_1$ | $C_2$ | $C_3$ | All | $D$ |
| TrialAttack | 0.0163 | 0.0013 | 0.0024 | 0.0095 | 2.91 | 0.0090 | 0.0191 | 0.0063 | 0.0135 | 6.26 | 0.0182 | 0.0391 | 0.0127 | 0.0201 | 6.17 |
| TNA | 0.0013 | 0.0002 | 0.0073 | 0.0023 | 2.83 | 0.0036 | 0.0119 | 0.0045 | 0.0081 | 5.32 | 0.0099 | 0.0359 | 0.0142 | 0.0181 | 5.28 |
| RevAdv | 0.1389 | 0.0013 | 0.0073 | 0.0758 | 2.32 | 0.0139 | 0.0410 | 0.0151 | 0.0282 | 5.59 | 0.0163 | 0.0609 | 0.0328 | 0.0353 | 5.97 |
| PAPU | 0.1138 | 0.0026 | 0.0066 | 0.0626 | 2.39 | 0.0223 | 0.0451 | 0.0165 | 0.0327 | 6.78 | 0.0216 | 0.0909 | 0.0375 | 0.0459 | 5.06 |
| DADA-DIV | **0.1966** | **0.0751** | **0.0267** | **0.1294** | **4.18** | **0.0577** | **0.0766** | **0.0336** | **0.0616** | **8.51** | **0.0405** | **0.1191** | **0.0704** | **0.0745** | **7.09** |

$\triangle \hat{H}R(C_j) = \frac{\triangle HR(C_j)}{\sum_{k=1}^{n_c} \triangle HR(C_k)}$. Then, the diversity brought by fake user behaviors can be measured based on the standard variance of normalized HR improvements as:

$$D(\mathbf{R}^f) = \frac{1}{\sqrt{\frac{\sum_{j=1}^{n_c}(\triangle \hat{H}R(C_j) - \triangle \hat{H}R_{mean})^2}{n_c}}}$$

where $\triangle HR_{mean} = \frac{\sum_{k=1}^{n_c} \triangle \hat{H}R(C_k)}{n_c}$ is the mean of the HR improvement of communities. A smaller diversity value indicates that the standard variance of normalized HR improvements of communities are higher, implying that fake user behaviors are dominated by larger communities.

To more comprehensively demonstrate the diversity-deficit issue, we compute the diversity value of existing attackers on three datasets. Specifically, we follow the same experiment setting in Sec. 3.2.1. We randomly select one item as the target item and separate real users into three communities by K-means based on the learned user representations. We use HR@50 as evaluate metric, and report the hit ratio improvement $\triangle HR$ on community 1, 2, 3, and all users, and the diversity value of each attacker. As shown in Tab. 10, under the same dataset, our proposed diversity-aware objective DADA-DIV outperforms baselines on each community and has the largest diversity value. It is because the diversity-aware objective more emphasizes the less affected users in each community when optimizing attackers. Thus, the generated fake user behaviors can manipulate the target RecSys to promote the target item to more users diversely, alleviating the diversity-deficit issue.

## B.8 Parameter Sensitivity

In this subsection, we evaluate DADA under different parameters. As shown in Tab 2 and Tab. 3, the results of DADA and baselines on different target RecSys and hit ratios have similar trends. For clarification, we use WRMF and HR@50 for evaluation.

**Extreme Limited Fake User Budget.** To better mimic the real-world applications, we set the user budget ratio $\beta_1$ as 0.001 and 0.0001. We take WRMF as the target RecSys and use the largest Gowalla dataset for evaluation. As illustrated in Tab. 11, our attacker outperforms these effective baselines under a limited budget, demonstrating its effectiveness. Besides, even under a fake user budget of 0.0001, our attacker DADA can achieve satisfactory performance, showing its practicability in real-world applications.

**Parameter Sensitivity on Hyperparamter $\lambda$.** We evaluate the effect of the trade-off parameter $\lambda$ between the difficulty-aware and the diversity-aware attack objective in Eq. (6). We vary $\lambda \in \{0, 10, 20, \cdots, 90, 100\}$, where higher $\lambda$ emphasizes more on diversity-aware attack objective. Overall, HR@50 on three datasets is quite robust to different $\lambda$ as shown in Fig. 7. More specifically, as $\lambda$ increases, the performance first increases, and then decreases. Furthermore, larger $\lambda$ will reduce the performance slightly. The reason is that larger $\lambda$ will force the attacker to concentrate more on improving each real user's latent tendency toward the target items. It implies that the attacker will treat real users as the same, and is unable to put more efforts on easy users who is simple to be affected, thereby reducing the attacker's effectiveness.

## B.9 Analysis on HR Improvement on the Target RecSys

In this subsection, we analyze the hit ratio (HR) improvement on different target RecSys. For more generality, we add a widely used RecSys ItemCF [35] as the target RecSys to evaluate our

Table 11: Evaluation on extreme limited fake user budget on the Gowalla dataset.

| Attacker | $|U^f| = 0.001 \cdot |U^r|$ | | | $|U^f| = 0.0001 \cdot |U^r|$ | | |
|---|---|---|---|---|---|---|
| | HR@10 | HR@50 | HR@100 | HR@10 | HR@50 | HR@100 |
| Clean | 0.0007 | 0.0052 | 0.0142 | 0.0007 | 0.0052 | 0.0142 |
| TrialAttack | 0.0007 | 0.0056 | 0.0147 | 0.0007 | 0.0055 | 0.0146 |
| TNA | 0.0007 | 0.0057 | 0.0146 | 0.0007 | 0.0052 | 0.0144 |
| RevAdv | 0.0008 | 0.0056 | 0.0149 | 0.0008 | 0.0056 | 0.0148 |
| PAPU | 0.0008 | 0.0061 | 0.0152 | 0.0008 | 0.0055 | 0.0149 |
| DADA | **0.0013** | **0.0072** | **0.0169** | **0.0009** | **0.0060** | **0.0154** |

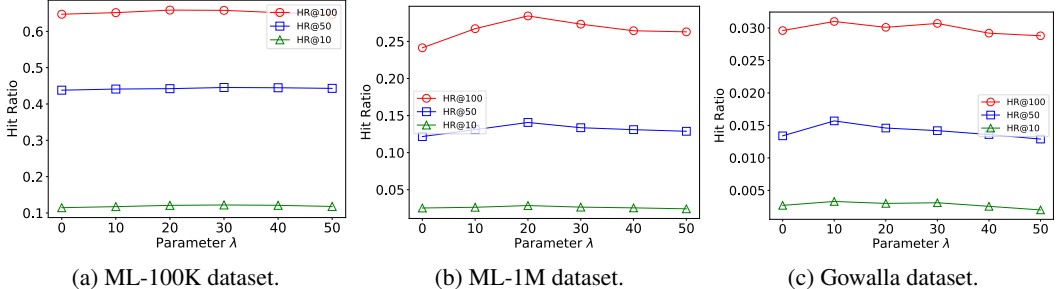

(a) ML-100K dataset.    (b) ML-1M dataset.    (c) Gowalla dataset.

Figure 7: Evaluation on the trade-off hyperparamter $\lambda$.

attacker. Formally, given the target RecSys $M_\theta$, we use $HR_{M_\theta}$ and $\hat{HR}_{M_\theta}$ to denote the hit ratio of target items on clean data and the data after injecting fake user behaviors, respectively. Under the target RecSys $M_\theta$, the HR improvement brought by fake user behaviors can be computed by $\hat{HR}_{M_\theta} - HR_{M_\theta}$. Specifically, we use HR@50 as evaluation metric and plot the HR improvement over the result in the clean data under different target RecSys.

As shown in Fig. 8, we can observe that if the hit ratio of a target RecSys on clean data is larger, the hit ratio improvement tends to be more significant, which is reasonable. It is because the higher hit ratio on clean data indicates that the target RecSys is willing to promote the target item to the top-N recommendation list of users. Thus, it is simpler to manipulate this target RecSys to promote the target item to more users. Consequently, the hit ratio improvement on the target RecSys is higher.

## B.10 Visualization

**Gradient Visualization.** Corresponding to Sec. 3.1.1, we show DADA's gradients and hit ratio on real users with different difficulties toward the target item. We follow the same experiment setting in Sec. 3.1.1. We visualize the gradients and HR@50 of the *easy*, *medium*, and *difficult* group. As shown in Fig. 9 (a), unlike existing attackers, such as RevAdv, which allow more difficult users to provide more gradients, our attacker allows easier users to contribute more gradients. Specifically, *easy* group contributes the most gradients, and *medium* group contributes more gradients than *difficult* group. It indicates our attacker put more effort into easier users. Correspondingly, as shown in Fig. 9 (b), the HR@50 of *easy* and *medium* groups increases significantly.

**Fake User Visualization.** Corresponding to Sec. 3.2.1, we visualize real and fake users based on the learned user representations and evaluate the hit ratio improvement in each community. Specifically, we follow the same experiment setting in Sec. 3.2.1. As shown in Fig. 10 (a), fake users are distributed diversely. Furthermore, the results on three datasets in Fig. 10 (b) (c) and (d) demonstrate that, unlike state-of-the-art attackers, such as PAPU and RevAdv, our attacker improves the HR@50 on three communities significantly. It is because our attacker allows unaffected real users from various communities to provide more gradients to the attacker's parameter optimization, avoiding large communities dominating the generated fake user behaviors.

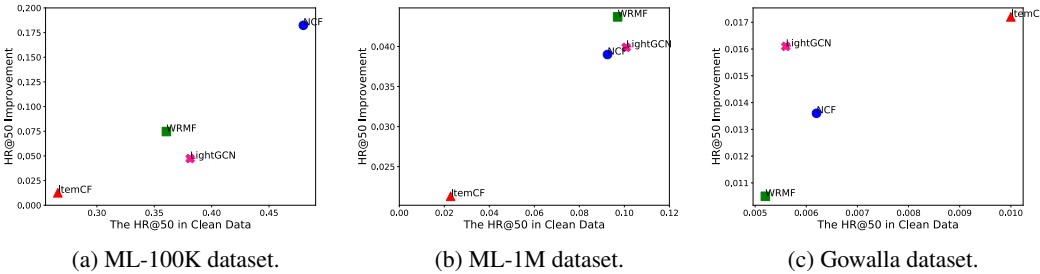

(a) ML-100K dataset.      (b) ML-1M dataset.      (c) Gowalla dataset.

Figure 8: Evaluation on HR improvement.

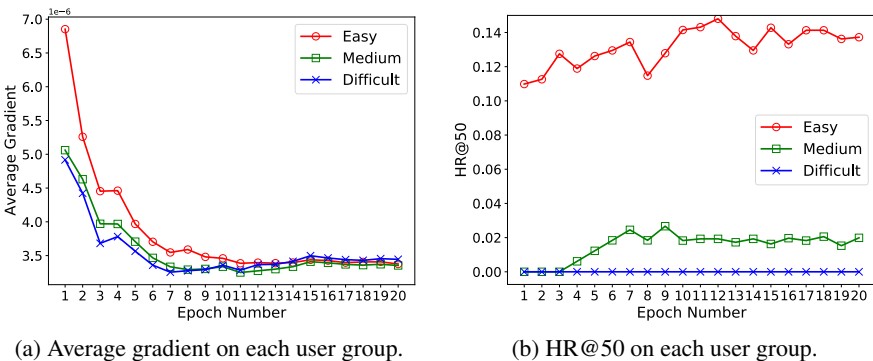

(a) Average gradient on each user group.      (b) HR@50 on each user group.

Figure 9: Visualization on gradients on ML-100K dataset.

## B.11 Future Directions

In this subsection, we introduce several future directions from the perspective of dynamic scenario, side information, and automated graph neural network.

- **Dynamic Scenario.** In real-world applications, the collected training user behaviors and the fake user budget may be changed. Thus, one promising future direction is how to efficiently create new fake behaviors based on the previous generation process instead of from scratch. One warm-up strategy is to fine tune the new fake behaviors based on the previously generated fake behaviors and new arriving real user behaviors. Besides, we can store the intermediate gradients of real users on optimizing the previous fake behaviors and then reuse them with the cache technique [13, 21] to accelerate new fake behaviors generation process.

- **Side Information.** Most existing injective attackers only employ real user behaviors to learn user and item representations, and then generate fake user behaviors accordingly. However, the fake user behaviors are sparse and only contain the user-item interactions, which hinders learning more expressive user and item representations. Thus, one promising direction is to incorporate side information to enhance the user behaviors, such as user social relationships [2] and numerous item information in knowledge bases [9, 25, 26, 37].

- **Automated Graph Neural Network.** As discussed before, graph neural networks (GNNs) have demonstrated their effectiveness in RecSys [48, 50]. More recently, automated GNNs [42, 43, 54, 55] propose to automatically design the best GNN architecture for any given data, which achieve SOTA performance on many graph-related tasks. However, there are few studies on the performance of those performance-driven GNNs under attack. It may be a more practical topic since the automated way may be deployed for recommendation purposes in real industries.

## B.12 Discussion on Defending Injective Attacks

In this paper, we first revisit the attack patterns of existing injective attackers and then propose a more effective difficulty-aware and diversity-aware attacker. In this subsection, we will discuss how the platforms effectively defend against injective attacks based on our observations.

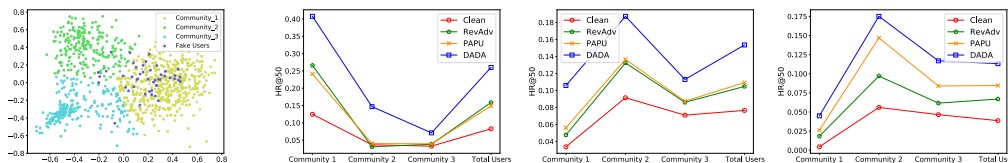

(a) Fake user distribution on ML_100k.

(b) HR@50 of each community ML_100k.

(c) HR@50 of each community on ML_1M.

(d) HR@50 of each community on Gowalla.

Figure 10: Visualization on community.

First, inspired by current noise reduction work [22, 23, 46], we can develop a fake user detector to identify fake users $U^f$ among all users $U$ in the platforms. Then, we can remove these potential fake users from the training data and only use behaviors of reliable users $U \setminus U^f$ to train the target RecSys $\mathcal{M}_\theta$. The process can be formulated as follows:

$$\theta^* = \arg \min_\theta \mathcal{L}_{tra}(\mathcal{M}_\theta(\mathbf{R}[U \setminus U^f]), \mathbf{R}[U \setminus U^f]),$$

Here we introduce two directions to develop a powerful fake user detector. (1) Current studies [7] have demonstrated that the item rating distributions of fake users are different from those of normal users. Thus, we can compute the rating derivation of each user from the mean value of all users. Then, users whose rating distribution significantly deviates from the mean distribution of all users can be considered to be fake users. (2) As shown in Fig. 1 (c) and Fig. 4 (a) and (c), the fake users generated by attackers tend to form a cluster. Thus, if we can identify some fake users in advance, we are able to identify the remaining fake users by examining users in the same cluster as the identified fake users. By these two ways, we can detect fake users and alleviate the negative impact of attacks on the target RecSys of platforms.

Second, adversarial training [53] is a popular and effective method for training a robust RecSys to resist attacks. The basic idea is to generate adversarial fake user behaviors by increasing the recommendation error of RecSys, and then use these adversarial user behaviors together with real user behaviors to optimize the RecSys parameters by minimizing the training loss. The critical part of such adversarial training is to create effective fake user behaviors to affect RecSys. In this paper, we have demonstrated that our proposed difficulty-aware and diversity-aware attacker DADA is capable of generating fake user behaviors that effectively affect the RecSys. Thus, our attacker DADA can be used to generate fake user behaviors to train the target RecSys in the adversarial manner. Specifically, given the target item set $I_t$ that attackers want to promote, we can generate fake user behaviors $\mathbf{R}^f$ with fake user budget $\delta$ by maximizing the attack objective $\mathcal{O}_{atk}(\cdot)$ in Eq. (6). Then, the target RecSys takes both fake user behaviors and real user behaviors as inputs to predict the tendency score for real users, i.e., $\mathcal{M}_\theta([\mathbf{R}^r, \mathbf{R}^f])[U^r] \in \mathbb{R}^{|U^r| \times |I|}$. The target RecSys can be optimized by minimizing the distance between predicted scores $\mathcal{M}_\theta([\mathbf{R}^r, \mathbf{R}^f])[U^r]$ and real user behaviors $\mathbf{R}^r$. This process can be formulated as follows:

$$\theta^* = \arg \min_\theta \mathcal{L}_{tra}(\mathcal{M}_\theta(\mathbf{R}^r), \mathbf{R}^r) + \lambda_{adv} \cdot \mathcal{L}_{tra}(\mathcal{M}_\theta([\mathbf{R}^r, \mathbf{R}^f])[U^r], \mathbf{R}^r),$$

$$s.t. \quad \mathbf{R}^f = \arg \max_{\mathbf{R}^f} \sum_{t \in I_t} \mathcal{O}_{atk}(\mathcal{M}_{\theta^*}(\mathbf{R}^r); U^r, t), \quad |U^f| \leq \delta,$$

where $\lambda_{atk}$ is a trade-off hyperparameter between normal training loss and adversarial loss. In future, we will utilize our proposed DADA to train a robust RecSys in the adversarial training manner.