# OpenReview forum: "Revisiting Injective Attacks on Recommender Systems"
_NeurIPS.cc/2022/Conference — NeurIPS 2022 Accept_

### Official Review · Reviewer_e81i · 2022-07-04

**Rating:** 7
**Confidence:** 3
**Soundness:** 3 good
**Presentation:** 3 good
**Contribution:** 3 good

**Summary:**

In this paper, the authors pointed out two issues with existing recommender system attackers, difficulty-agnostic and diversity-deficit. They then analyzed these two issues and proposed two ways to alleviate them. Both methods modify the loss function and introduced new notions in the gradient. Comparison with multiple baselines over the ML-100K and ML-1M datasets showed that the proposed approach can outperform all of them.

**Questions:**

Please see my questions in the last section.

**Limitations:**

It would be great and super helpful to have more discussions (with more details) on the defense of the proposed attack.

**Strengths And Weaknesses:**

Strength:
1. The diversity-agnostic issue is well explained and justified. The proposed solution is intuitive and reasonable.
2. The authors conducted comparisons with more than 10 baselines and did very detailed analysis.
3. Different recommendation algorithms (WRMF, NCF, LightGCN) were experimented with, which demonstrates the generalisability and robustness of the proposed approach.

Weakness:
1. The diversity-deficit issue is not very convincing to me. Do the fake injected users always form a cluster that is always a subgroup of a larger cluster of users? More explanations and proof (if there is any) of this phenomenon can be very helpful to the readers.
2. I wonder if the choice of \beta_1 and \beta_2 can reflect the real-world scenario. In other words, is it possible to inject 1% of users with 1% of items into a recommender system in real world? This seems to be quite unlikely. It would be very interesting to see the results when \beta_1 and \beta_2 are much smaller (0.01%ish), which can mimic the real-world scenario better.
3. It would be helpful to annotate if the results in Table 1 and 2 shows significant differences.
4. It would be great and super helpful to have more discussions (with more details) on the defense of the proposed attack.

---

> ### Author Response · Authors · 2022-08-02
> **Response to reviewer e81i, including more discussions and more experiments**
>
> ## Replies to Weaknesses
>
> **e81i-W1:** The diversity-deficit issue needs more clarification. Specifically, in the diversity-deficit issue, do the injected fake users always form a cluster that is always a subgroup of a larger cluster of users? More explanations and proof (if there is any) of this phenomenon can be very helpful to the readers.
>
> **Rely to e81i-W1**:  Thanks for your suggestions.
> Here we first clarify the concept of the diversity-deficit issue and then explain the reason why fake user behaviors generated by existing attackers tend to form a cluster. We have included this discussion in Sec. 3.2.1.
>
>
> (1). The diversity-deficit issue indicates that the fake behaviors generated by attackers are dominated by large communities. As a result, these fake user behaviors can only manipulate the target RecSys to promote the target item to the dominated communities. In other words, the target RecSys is incapable of promoting the target item to more users in other communities diversely. This phenomenon can be observed in Fig. 1 (d) in Sec. 3.2.1 and Fig. 3 (b)(d)(f) in Appx. A.1.2, i.e., the generated fake users only improve the HR@50 of one community significantly while maintaining similar HR@50 in the other two communities.
>
> (2). If the attacker does not force the generated fake users to affect users diversely,  the fake users tend to form a cluster that is dominated by large communities. Specifically, as introduced in Eq. (3), each real user $u\in U^r$ will provide gradients to optimize the attackers' parameters $\varphi$.
> Since similar real users have similar preferences on items, real users can form several communities $\\{C_j\\}\_{j=1}^{n_c}$, where each community $C_j$ has $n_j$ real users.
> From the community perspective, the optimization process of the attacker parameter $\varphi$  via maximizing the attack objective $\mathcal{O}\_{atk}(\cdot)$ can be reformulated as follows:
>
> $$
> \varphi=\varphi + \alpha \cdot \sum_{j=1}^{n_c}{ \sum_{u \in C_j}{\frac{\partial \mathcal{O}\_{atk}(\mathcal{{M}}\_{\theta^\*}(\mathbf{R});u,t)}{\partial \varphi}}}
> $$
>
> Intuitively, users in each community who are similar will give the gradients $\frac{\partial \mathcal{O}\_{atk}(\mathcal{M}\_{\theta^\*}(\mathbf{R});u,t)}{\partial \varphi}$ in similar directions. Thus, there is a potential risk that the large communities $C_k$ will dominate the optimization direction of  attacker parameter $\varphi$ via $\sum_{u \in C_k}{\frac{\partial \mathcal{O}\_{atk}(\mathcal{{M}\}\_{\theta^\*}(\mathbf{R});u,t)}{\partial \varphi}}$.
> Consequently, the generated fake user behaviors will be dominated by large communities and form a cluster. One the other hand, from the Fig. 1(c) in Sec. 3.2.1 and Fig. 3(a)(c) in Appx. A.1.2, we can observe that fake users tend to form a cluster.

---

> > ### Author Response · Authors · 2022-08-02
> > **Response to weakness 2 and 3**
> >
> >
> > **e81i-W2:** Whether the choice of $\beta_1$ and $\beta_2$ can reflect the real-world scenario. It would be very interesting to see the results when $\beta_1$ and $\beta_2$ are much smaller, such as 0.01%, which can mimic the real-world scenario better.
> >
> > **Rely to e81i-W2:**  Thanks for your suggestions. In our paper, we use $\beta_1$ to control the number of injected fake users $\tau=\beta_1 \cdot |U^r|$, and use $\beta_2$ to control the maximum number of selected items  $\tau=\beta_2 \cdot |I|$. In the main experiments, we set the $\beta_1=0.01$ and set $\beta_2=\\{0.05,0.05,0.002\\}$ for three datasets as the default. Also, in the last version, we have conducted parameter sensitivity experiments in Appx. B.8 by varying $\beta_1, \beta_2 \in \\{0.01,0.03,0.05,0.07,0.09\\}$ on ML-100K and ML-1M datasets.
> > Intuitively, since platforms may have millions of users, it is realistic but costly to employ more than ten thousand users to create user behaviors to promote the target item. Besides, the $\beta_2$ setting for three datasets is reasonable, which is similar to the average number of interacted items of real users, i.e., $\\{0.063,0.045,0.002\\}$ for ML-100K, ML-100M, and Gowalla shown in Tab. 4. Actually, there is no need to set $\beta_2$ very small, since it is better to mimic the real user behaviors to avoid fake user detection from platforms.
> >
> > To better mimic the real-world applications, we set the user budget ratio $\beta_1$ as 0.001 and 0.0001. We take WRMF as the target RecSys and use the largest Gowalla dataset for evaluation. We report the results of HR@10/HR@50/HR@100 as follows:
> >
> > |    model    |         $\beta_1=0.001$         |        $\beta_1=0.0001$         |
> > | :---------: | :-----------------------------: | :-----------------------------: |
> > |    Clean    |      0.0007/0.0052/0.0142       |      0.0007/0.0052/0.0142       |
> > | TrialAttack |      0.0007/0.0056/0.0147       |      0.0007/0.0055/0.0146       |
> > |     TNA     |      0.0007/0.0057/0.0146       |      0.0007/0.0052/0.0144       |
> > |   RevAdv    |      0.0008/0.0056/0.0149       |      0.0008/0.0056/0.0148       |
> > |    PAPU     |      0.0008/0.0061/0.0152       |      0.0008/0.0055/0.0149       |
> > |    DADA     | $\textbf{0.0013/0.0072/0.0169}$ | $\textbf{0.0009/0.0060/0.0154}$ |
> >
> > As illustrated in the above table, our attacker outperforms these effective baselines under a limited budget, demonstrating its effectiveness. Besides, even under a fake user budget of 0.0001, our attacker DADA can achieve satisfactory performance, showing its practicability in real-world applications. We have included this experiments in  Appx. B.8.
> >
> > **e81i-W3:** It would be helpful to annotate if the results in Table 1 and Table 2 show significant differences.
> >
> > **Rely to e81i-W3:**  Thanks for your suggestions. We have annotated the improved HR ratio over clean data to show the significant difference of our attacker against baselines.

---

> > > ### Author Response · Authors · 2022-08-02
> > > **Response weakness 4 and limitation 1**
> > >
> > > ## Replies to Limitations
> > >
> > > **e81i-W4 and e81i-L1:** It would be great and super helpful to have more discussions (with more details) on the defense of the proposed attack.
> > >
> > > **Rely to e81i-W4 and e81i-L1:**  Thanks for your suggestions. In Appx. B.12, we have added more details on how to help the target RecSys in the platforms to defense against such injective attacks. Specifically, we introduce two approaches, i.e., fake user detection and adversarial training, to protect the platforms from injective attackers. Particularly, in the adversarial training approach, our proposed attacker can generate effective fake user behaviors to adversarially train a more robust RecSys for platforms. The details are as follows:
> > >
> > > First, we can develop a fake user detector to identify fake users $U^f$ among all users $U$ in the platforms. Then, we can remove these potential fake users from the training data and only use behaviors of reliable users $U\setminus U^f$  to train the target RecSys $\mathcal{M}\_{\theta}$. The process can be formulated as follows:
> > >
> > > $$
> > > \theta^\* = \arg\min_{\theta}{\mathcal{L}\_{tra}(\mathcal{M}\_{\theta}(\mathbf{R}\[U\setminus U^f\]),\mathbf{R}\[U\setminus U^f\])},
> > > $$
> > >
> > > Here we introduce two directions to develop a powerful fake user detector. (1) Current studies (eg. Sec. 3 in \[1\]) have demonstrated that the item rating distributions of fake users are different from those of normal users. Thus, we can compute the rating derivation of each user from the mean value of all users. Then, users whose rating distribution significantly deviates  from the mean distribution of all users can be considered to be fake users. (2) As shown in Fig.1(c) and Fig.3(a)(c), the fake users generated by attackers tend to form a cluster. Thus, if we can identify some fake users in advance, we are able to identify the remaining fake users by examining users in the same cluster. By these two ways, we can detect fake users and alleviate the negative impact of attacks on the target RecSys of platforms.
> > >
> > > Second, adversarial training is a popular and effective method for training a robust RecSys to resist attacks. The basic idea is to generate adversarial fake user behaviors by increasing the recommendation error of RecSys,
> > > and then use these adversarial user behaviors together with real user behaviors to optimize the RecSys parameters by minimizing the training loss.  The critical part of such adversarial training is to create effective fake user behaviors to affect RecSys. In this paper, we have demonstrated that our proposed difficulty-aware and diversity-aware attacker DADA is capable of generating fake user behaviors that effectively affect the RecSys.  Thus, our attacker DADA can be used to generate fake user behaviors to train the target RecSys in the adversarial manner. Specifically, given the target item set $I_t$ that attackers want to promote, we can generate fake user behaviors $\mathbf{R}^f$ with fake user budget $\delta$ by maximizing the attack objective $\mathcal{O}\_{atk}(\cdot)$ in Eq. (6).  Then, the target RecSys takes both fake user behaviors and real user behaviors as inputs to predict the tendency score for real users, i.e., $\mathcal{M}\_{\theta}(\[\mathbf{R}^r, \mathbf{R}^f\])\[U^r\]\in \mathbb{R}^{|U^r| \times |I|}$. The target RecSys can be optimized by minimizing the distance between predicted scores $\mathcal{M}\_{\theta}(\[\mathbf{R}^r, \mathbf{R}^f\])\[U^r\]$ and real user behaviors $\mathbf{R}^r$. This process can be formulated as follows:
> > >
> > > $$
> > > \begin{align}
> > > &	\theta^* = arg\min_{\theta}{\mathcal{L}\_{tra}(\mathcal{M}\_{\theta}(\mathbf{R}^r),\mathbf{R}^r)+\lambda_{adv}\cdot \mathcal{L}\_{tra}( \mathcal{M}\_{\theta}(\[\mathbf{R}^r, \mathbf{R}^f\])\[U^r\],\mathbf{R}^r)}, \\\\
> > > 	s.t. &\\ \mathbf{R}^f = arg\max_{\mathbf{R}^f}{\sum_{t \in I_t}{\mathcal{O}\_{atk}(\mathcal{{M}}\_{\theta^\*}(\mathbf{R}^r);U^r,t)} }, \quad |U^f| \le \delta,\nonumber
> > > \end{align}
> > > $$
> > >
> > > where $\lambda\_{atk}$ is a trade-off hyperparameter between normal training loss and adversarial loss.  In the future, we will utilize our proposed attacker DADA to train a robust RecSys in the adversarial training manner.
> > >
> > > \[1\] Davoudi A, Chatterjee M. Detection of profile injection attacks in social recommender systems using outlier analysis[C]//2017 IEEE International Conference on Big Data (Big Data). IEEE, 2017: 2714-2719.

---

> ### Comment · Reviewer_e81i · 2022-08-10
> **Thank you for the response**
>
> Thanks to the authors for their careful responses and revisions. Most of my questions have been answered. Overall I think this is a good paper worth publishing. After re-evaluation, I'd like to raise my rating accordingly.

---

### Official Review · Reviewer_Es2b · 2022-07-11

**Rating:** 7
**Confidence:** 3
**Ethics Flag:** Yes
**Soundness:** 3 good
**Presentation:** 3 good
**Contribution:** 3 good

**Summary:**

This paper pointed out two issues of existing injective attacks against recommender systems, difficulty-agnostic and diversity-deficit issues.
To address these two issues, the paper propose a difficulty and diversity aware attacker, name DADA, to force the attacker to attack easy users distributed in various communities.
The experimental results on three real-world datasets demonstrate that the proposed method achieves superior performance compared with state-of-the-art attackers.
Besides, the analytical experiments also show the proposed objectives could generated fake users indeed focus on easy users in a diverse way.


**Questions:**

see above

**Ethics Review Area:**

["Privacy and Security (e.g., consent)"]

**Limitations:**

The method proposed in this paper could be used to attack the real-world recommendation application for profits.

**Strengths And Weaknesses:**

## Strengths
* This is a well-structured paper with clear presentation.
* Motivational observation subsections are very insightful.
* The paper conducted very solid experiments with state-of-the-art baselines on three benchmark datasets.


## weakness
*  The paper lacks explanations for some phenomena in the experiments. For example, the results in Table 2 and Table 3 imply that stronger models are more robust to attacks.
*  Hit Ration ignore the rank of the target items in top-N recommended list. Thus, metrics like NDCG should be also reported in the experiments.

---

> ### Author Response · Authors · 2022-08-02
> **Response to weakness, including more discussions and more experiments.**
>
> We sincerely thank the reviewer’s efforts in reviewing our submission, and our detailed replies are as follows:
>
> ## Replies to Weaknesses
>
> **Es2b-W1:** The results in Table 2 and Table 3 imply that stronger target RecSys are more robust to attacks. It should add an explanation for this phenomenon.
>
> **Rely to Es2b-W1**:  Thanks for pointing it out. To further investigate this phenomenon, we analyze the HR Improvement on different target RecSys in Appx. B.9.
>
> For more generality, we add  ItemCF\[1\] as the target RecSys to evaluate our attacker.
> Formally, given the target RecSys $M_\theta$, we use $HR\_{M_\theta}$ and $\hat{HR}\_{M_\theta}$ to denote the hit ratio of target items on clean data and the data after injecting fake user behaviors, respectively. Under the target RecSys $M_\theta$, the HR improvement brought by fake user behaviors can be computed by  $\hat{HR}\_{M_\theta} - HR\_{M_\theta}$.
> Specifically, we use HR@50 as evaluation metric and plot the HR improvement over the result in the clean data under different target RecSys.
>
> As shown in Fig. 8 in Appx. B.9, we can observe that if the hit ratio of a target RecSys on clean data is larger, the hit ratio improvement tends to be more significant. Intuitively, from the attack perspective, the target RecSys is stronger if the hit ratio of the target items on clean data is smaller. The results indicate that stronger RecSys are more robust to attacks.
> It is because the weaker target RecSys that have a higher hit ratio on clean data are more willing to promote the target item to the top-K recommendation list of users.
> Thus, it is simpler to manipulate such target RecSys to promote the target item to more users. Consequently, the hit ratio improvement on the weaker target RecSys is larger.
>
> \[1\]Sarwar, Badrul, et al. "Item-based collaborative filtering recommendation algorithms." WWW2001
>
>
>
>
> **Es2b-W2**: Hit Ratio ignores the rank of the target items in the top-N recommended list. Thus, metrics like NDCG should also be reported in the experiments.
>
> **Rely to Es2b-W2**:  Thanks for your suggestions. We have utilized the metric NDCG to evaluate the ranking quality of target items in the recommendation list. In particular, we report the results NDCG@50 of our attacker and baselines on three datasets, since results on NDCG@10 and NDCG@100 have a similar trend with NDCG@50. We list the results on three datasets evaluated by the target RecSys WRMF as follows, and more evaluations on NCF and LightGCN have been added in Tab. 7 in Appx. B.4.
>
> |   Dataset   |       ML-100k        |        ML-1M         |       Gowalla        |
> | :---------: | :------------------: | :------------------: | :------------------: |
> |    Clean    |        0.0450        |        0.0089        |        0.0004        |
> |   Random    |        0.0447        |        0.0099        |        0.0005        |
> |   Popular   |        0.0453        |        0.0100        |        0.0006        |
> |    CoVis    |        0.0458        |        0.0101        |        0.0008        |
> | TrialAttack |        0.0478        |        0.0097        |        0.0006        |
> |    SRWA     |        0.0471        |        0.0094        |        0.0006        |
> |     PGA     |        0.0464        |        0.0100        |        0.0006        |
> |    SGLD     |        0.0469        |        0.0095        |        0.0006        |
> |     TNA     |        0.0476        |        0.0103        |        0.0009        |
> |   RevAdv    |        0.0560        |        0.0107        |        0.0016        |
> |    PAPU     |        0.0566        |        0.0109        |        0.0018        |
> |  DADA-DICT  | $\underline{0.0581}$ | $\underline{0.0131}$ | $\underline{0.0028}$ |
> |  DADA-DIV   |        0.0541        |        0.0114        |        0.0023        |
> |    DADA     |  $\textbf{0.0612}$   |  $\textbf{0.0136}$   |  $\textbf{0.0032}$   |
>
>
> As shown in the above table, heuristics-based attackers,
> as Random, Popular, and CoVis,  achieve unsatisfied results on NDCG@50. It is because they cannot optimize the attack objective directly. Besides, compared with TrialAttack, RevAdv, and PAPU, several gradient-based attackers, including SRWA, PGA, SGLD, and TNA, achieve inferior performance on NDCG. It is because they compute the gradients in a biased way, thus damaging the rank quality of target items. Finally, our proposed attacker DADA achieves the best performance than baselines, demonstrating that the rank quality of target items is high.

---

> ### Author Response · Authors · 2022-08-02
> **Response to limitation**
>
>
> ## Replies to Limitations
>
> **Es2b-L1:** The method proposed in this paper could be used to attack the real-world recommendation application for profits.
>
> **Rely to Es2b-L1**: Thanks for your comments. In Appx. B.12, we have added more details on how to help the platforms resist injective attacks. Specifically, inspired by our work, we introduce two approaches, i.e., fake user detection and adversarial training. Particularly, in the adversarial training approach, our proposed attacker can generate effective fake user behaviors to adversarially train a more robust RecSys for platforms. The details are as follows:
>
> First, we can develop a fake user detector to identify fake users $U^f$ among all users $U$ in the platforms. Then, we can remove these potential fake users from the training data and only use behaviors of reliable users $U\setminus U^f$  to train the target RecSys $\mathcal{M}\_{\theta}$. The process can be formulated as follows:
>
> $$
> \theta^\* = \arg\min_{\theta}{\mathcal{L}\_{tra}(\mathcal{M}\_{\theta}(\mathbf{R}\[U\setminus U^f\]),\mathbf{R}\[U\setminus U^f\])},
> $$
>
> Here we introduce two directions to develop a powerful fake user detector. (1) Current studies (e.g.,Sec. 3 in \[1\]) have demonstrated that the item rating distributions of fake users are different from those of normal users. Thus, we can compute the rating derivation of each user from the mean value of all users. Then, users whose rating distribution significantly deviates  from the mean distribution of all users can be considered to be fake users. (2) As shown in Fig.1(c) and Fig.3(a)(c), the fake users generated by attackers tend to form a cluster. Thus, if we can identify some fake users in advance, we are able to identify the remaining fake users by examining users in the same cluster. By these two ways, we can detect fake users and alleviate the negative impact of attacks on the target RecSys of platforms.
>
> Second, adversarial training is a popular and effective method for training a robust RecSys to resist attacks. The basic idea is to generate adversarial fake user behaviors by increasing the recommendation error of RecSys,
> and then use these adversarial user behaviors together with real user behaviors to optimize the RecSys parameters by minimizing the training loss.  The critical part of such adversarial training is to create effective fake user behaviors to affect RecSys. In this paper, we have demonstrated that our proposed difficulty-aware and diversity-aware attacker DADA is capable of generating fake user behaviors that effectively affect the RecSys.  Thus, our attacker DADA can be used to generate fake user behaviors to train the target RecSys in the adversarial manner. Specifically, given the target item set $I_t$ that attackers want to promote, we can generate fake user behaviors $\mathbf{R}^f$ with fake user budget $\delta$ by maximizing the attack objective $\mathcal{O}\_{atk}(\cdot)$ in Eq. (6).  Then, the target RecSys takes both fake user behaviors and real user behaviors as inputs to predict the tendency score for real users, i.e., $\mathcal{M}\_{\theta}(\[\mathbf{R}^r, \mathbf{R}^f\])\[U^r\]\in \mathbb{R}^{|U^r| \times |I|}$. The target RecSys can be optimized by minimizing the distance between predicted scores $\mathcal{M}\_{\theta}(\[\mathbf{R}^r, \mathbf{R}^f\])\[U^r\]$ and real user behaviors $\mathbf{R}^r$. This process can be formulated as follows:
>
> $$
> \begin{align}
> &	\theta^* = arg\min_{\theta}{\mathcal{L}\_{tra}(\mathcal{M}\_{\theta}(\mathbf{R}^r),\mathbf{R}^r)+\lambda_{adv}\cdot \mathcal{L}\_{tra}( \mathcal{M}\_{\theta}(\[\mathbf{R}^r, \mathbf{R}^f\])\[U^r\],\mathbf{R}^r)}, \\\\
> 	s.t.\\  & \mathbf{R}^f = arg\max_{\mathbf{R}^f}{\sum_{t \in I_t}{\mathcal{O}\_{atk}(\mathcal{{M}}\_{\theta^\*}(\mathbf{R}^r);U^r,t)} }, \quad |U^f| \le \delta,
> \end{align}
> $$
>
> where $\lambda\_{atk}$ is a trade-off hyperparameter between normal training loss and adversarial loss.  In the future, we will utilize our proposed attacker DADA to train a robust RecSys in the adversarial training manner.
>
>
> \[1\] Davoudi A, Chatterjee M. Detection of profile injection attacks in social recommender systems using outlier analysis[C]//2017 IEEE International Conference on Big Data (Big Data). IEEE, 2017: 2714-2719.

---

### Official Review · Reviewer_CMWp · 2022-07-25

**Rating:** 5
**Confidence:** 3
**Soundness:** 3 good
**Presentation:** 2 fair
**Contribution:** 3 good

**Summary:**

The paper investigates injection attacks on recommender systems, where an adversary injects a number of fake user profiles into the training data of the system, with behaviors chosen such that the system is more likely to recommend specific items to real users. To this end, the authors investigate existing attack schemes and find that they spend too much effort -- and thus fake-user budget -- on targeting users that are hard to recommend the chosen item to (i.e. the attacks are difficulty-agnostic) and that they predominantly target the chosen item to users belonging to a specific cluster of the population (i.e. they suffer from a diversity deficit).
To mitigate these issues, the authors propose two new targeting objectives: a difficulty-aware objective that focuses the optimization effort on users closer to the top-k threshold of having the target item recommended and a diversity-aware objective that steers the optimization towards users where it can cause the largest increase in recommendation score. These objectives are then combined to create a new type of attacker called DADA (difficulty-aware and diversity-aware), together with a greedy algorithm that optimizes the joint objective. In an empirical evaluation on three datasets with three types of recommender systems, the authors demonstrate that DADA outperforms existing attacks, leading to a higher number of users that receive the target item(s) in their top-k recommendations.

**Questions:**

- Why does the diversity-aware objective increase diversity, i.e. why did you chose it to be what it is? As far as I understand, the objective seems to primarily focus optimization effort on users where this effort would maximize the increase in targeting score for the target item. This in itself seems reasonable, but I am not convinced that it will necessarily increase the diversity in the targeted set of users. There also seems to be no metric that quantitatively measures diversity in the paper, so the provided visual results (scatter plots) feel a inconclusive regarding a lack of diversity for existing methods as well as an increase in diversity as the result of applying the diversity-aware objective. To be clear: I do not necessarily think that the objective is a bad one, but I don't think it can be called diversity-aware with the current amount of justification and doing so can be misleading.
- Were the experiments conducted over multiple runs? It looks like there is only one run (see weaknesses).
- As an ablation, it might be interesting to study some of the existing objectives (e.g. RevAdv) with the change of zeroing gradients corresponding to users where the target item is in the top-k recommendations. This might reveal how much of the gains come from the changed objectives, vs how much from concentrating optimization effort on users where the goal is not reached yet.
- The description of the attack objectives describes the case where there is one target item, however, in the empirical analysis target sets of multiple items are used. In this case, do you zero-out the contribution of a user to the gradient as soon as one of the target items appears in the top-k recommendations or do you continue until all items are recommended?
- Why is the difficulty-aware objective called $Ten(.)$?

**Limitations:**

- The proposed attack might run into some scalability issues, since for any change in the training dataset or for generating more or less fake users a full re-run is required. Since this is an attack and not a production system it is probably ok, but it might be helpful to briefly mention this and maybe discuss whether there is the possibility of a warm-start using a previously computed solution.
- Again, the threat-model should be discussed more extensively in this context, in particular which types of platforms offer enough public data to enable the attacks.

**Strengths And Weaknesses:**

### Strengths
- The paper proposes two interesting hypothesis for why existing injection attacks on recommender systems can fail to be effective in promoting items to more users: a) being difficulty-agnostic and wasting attack-budget on users where it likely won't make a difference instead of succeeding for more easier users and b) suffering from a lack of diversity in in the set of targeted users and only reaching sub-communities.
- The paper introduces two new targeting objectives as well as a way to combine them to mitigate these problems. It empirically demonstrates that these objectives lead to more effective attacks where the target item is recommended to more users, outperforming prior approaches.
- The paper is overall easy to understand and follow, though there are a number of presentation issues (see weaknesses).

### Weaknesses
- It is not clear how the diversity-aware objective increases diversity, and more justification is needed. See the questions part for more details.
- More clarity is needed regarding the threat-model. The paper assumes a gray-box threat-model where attackers don't have access to the targeted system, but have knowledge of the user data used to train the system such as movie viewing history. This seems like a strong assumption to me, though I'm willing to accept it, given that prior work also makes it. However, there needs to be more discussion of how this training data can be obtained, i.e. which pieces of information about user behavior are typically accessible on platforms and whether they are sufficient to train a surrogate model.
- As far as I can see and inferring from the review-checklist, the results (e.g. in table 3 and 4, figure 1) are only computed for one run. Given that there are a number of stochastic components involved here (the selection of the target items, k-means initialization, etc.), results should really be provided as aggregates over multiple runs, else the confidence in the conclusions is low. If the based on multiple runs this should be clearly stated and e.g. error bars should be provided.
- There are some presentation issues:
    - I have an issue with some of the formulations repeatedly used throughout the paper, for instance:
        - The phrase "without loss of generality" is used a few times (e.g. in section 3.1.1 and 3.2.1) to refer to some of the choices made in the experiments. However, in its precise use, this phrase is reserved for assumptions in mathematical statements and not to justify design choices in empirical experiments. If you would want to use it there, you should show that other choices lead to similar results.
        - The phrase some "users contribute more gradients" is used a number of times. It's possible to understand it's meaning, but it feels rather imprecise. It could be understood as some users appearing multiple times in the objective and therefore being counted more than once in the gradient computation. An alternative such as some "users dominate the gradient computation" or something similar seems preferable.
    - There are a number of language and grammatical issues that make the paper feel not very polished.

---

> ### Author Response · Authors · 2022-08-02
> **Response to Reviewer CMWp, including more discussions and experiments**
>
> We sincerely thank the reviewer’s efforts in reviewing our submission, and our detailed replies are as follows:
>
> ## Replies to Weaknesses
>
> **CMWp-W1 and CMWp-Q1:** The presentation of the diversity-aware objective is not clear. (1) Why existing attackers have the $\textit{diversity-deficit}$ issue, and why the diversity-aware objective can enable the proposed attacker to create fake user behaviors that can affect real users in a diverse manner? (2) How to quantitatively measure diversity?
>
> **Reply to CMWp-W1 and CMWp-Q1:** Thanks for pointing it out. Due to the space limit, we did not elaborate too much on the diversity-aware objective. But we have realized that the description of the diversity-aware objective is insufficient and will include the following discussion of question (1) in Sec. 3.2 and the discussion of question (2) in Appx. B.7.
>
> (1) Generally, $\textit{diversity-deficit}$ issue is that the generated fake behaviors of attackers are dominated by large communities, where users in each community have similar preferences. The reason is that users in the same community who are similar will give the gradients $\frac{\partial \mathcal{O}\_{atk}(\mathcal{{M}}\_{\theta^\*}(\mathbf{R});u,t)}{\partial \varphi}$ in similar directions. Thus, in existing attackers, there is a potential risk that the large communities $C_k$ will dominate the optimization direction of  attacker parameter $\varphi$ via $\sum_{u \in C_k}{\frac{\partial \mathcal{O}\_{atk}(\mathcal{M}\_{\theta^\*}(\mathbf{R});u,t)}{\partial \varphi}}$.
> As a result, the fake user behaviors generated by attackers can only manipulate the target RecSys to promote the target item to the dominated communities. In other words, the target RecSys is incapable of promoting the target item to more users in other communities diversely. This phenomenon can be observed clearly in Fig. 1(d) in Sec. 3.2.1 and Fig. 3(b)(d)(f) in Appx. A.1.2, i.e., the generated fake users only improve the HR@50 of one community significantly while maintaining similar HR@50 in the other two communities.
>
> To address the diversity-deficit issue, if the real users are less influenced by fake user behaviors, our diversity-aware objective in Eq. (5) will more focus on these real users when optimizing the attack parameters.  The influence of fake users for real user $u$ can be measured by $\triangle \mathbf{\hat{S}}\[u\]\[t\]=\mathbf{\hat{S}}\[u\]\[t\] - \mathbf{\hat{S}'}\[u\]\[t\]$ is the tendency difference on the target item $t$ after injecting fake users.
> **In this way, if users of one community are not affected by fake users, the diversity-aware objective will increase the weight of these users in the subsequent gradient computation. Then, the attacker will be optimized to generate fake user behaviors that can affect these users. Thus, the diversity-aware objective enables fake user behaviors generated by our attacker to affect real users diversely.**
>
> Besides, we add two more experiments on ML-1M and Gowalla datasets in Fig. 10 (c) and (d) in Appx. B.10 to show the hit ratio improvement of each community against existing attackers. The results show that, unlike existing attackers that focus on one community, our attacker can improve the HR ratio of all communities simultaneously. It indicates that our attacker enables the target RecSys to promote the target item to users in different communities diversely, addressing the diversity-deficit issue.

---

> > ### Author Response · Authors · 2022-08-02
> > **Response to the second question of weakness 1 (part 2)**
> >
> > (2) To address the $\textit{diversity-deficit}$ issue, we expect that fake user behaviors created by attackers that can manipulate the target RecSys to recommend the target into the top-k list of users in different communities. Based on this, we can quantitatively measure the diversity brought by each attacker based on the hit ratio improvement of each community. Formally, given communities $\\{C_j\\}\_{j=1}^{n_c}$ and a target item $t$, we denote the hit ratio improvement of each community $C_j$ after injecting fake  behaviors as $\triangle HR(C_j)$, and the normalized hit ratio difference can be computed as $\triangle \hat{HR}(C_j)= \frac{\triangle HR(C_j)}{\sum_{k=1}^{n_c}{\triangle HR(C_k)}}$. Then, the diversity brought by fake user behaviors  $\mathbf{R}^f$ can be measured based on the standard variance of normalized HR improvements as:
> >
> > $$
> > D(\mathbf{R}^f)=\frac{1}{\sqrt{\frac{\sum_{j=1}^{n_c}{(\triangle \hat{HR}(C_j)-\triangle \hat{HR}\_{mean})^2}}{n_c}}}
> > $$
> >
> > where $\triangle HR_{mean} = \frac{\sum_{k=1}^{n_c}{\triangle \hat{HR}(C_k)}}{n_c}$ is the mean of the HR improvement of communities. A smaller diversity value indicates that the standard variance of normalized HR improvements of communities is higher, implying that fake user behaviors are dominated by larger communities.
> >
> > To more comprehensively demonstrate the diversity-deficit issue, we compute the diversity value of existing attackers on three datasets. For simplicity and clarification, we randomly select one item as the target item and separate real users into three communities by K-means based on the learned user representations. We use HR@50 as the evaluation metric, and report the hit ratio improvement $\triangle HR$ on community 1/2/3/all users and the diversity value of each attacker as follows:
> >
> >
> >
> >
> > |    model    | ML-100k||||| ML-1M|||||
> > | :---------: | :-----: | :---: | :-----: | :---------: | :-----: | :---: | :-----: | :---------: | :-----: | :---: |
> > |-|$\triangle HR(C_1)$|$\triangle HR(C_2)$|$\triangle HR(C_3)$|All|$D(\mathbf{R}^f)$|$\triangle HR(C_1)$|$\triangle HR(C_2)$|$\triangle HR(C_3)$|All|$D(\mathbf{R}^f)$|
> > |      TrialAttack     | 0.0163|0.0013|0.0024|0.0095|2.91   |   0.0090|0.0191|0.0063|0.0135|6.26|
> > |       TNA     |  0.0013|0.0002|0.0073|0.0023|2.83  |  0.0036|0.0119|0.0045|0.0081 | 5.32  |
> > |RevAdv|0.1389|0.0013|0.0073|0.0758|2.32|0.0139|0.0410|0.0151|0.0282|5.59|
> > |PAPU|0.1138|0.0026|0.0066|0.0626|2.39|0.0223|0.0451|0.0165|0.0327|6.78|
> > |DADA-DIV|$\textbf{0.1966}$|$\textbf{0.0751}$|$\textbf{0.0267}$|$\textbf{0.1294}$|$\textbf{4.18}$ |$\textbf{0.0577}$|$\textbf{0.0766}$|$\textbf{0.0336}$|$\textbf{0.0616}$|$\textbf{8.51}$|
> >
> >
> > |    model    | Gowalla |||||
> > | :---------: | :-----: | :---------: | :-----: | :---: | :-----: |
> > |-|$\triangle HR(C_1)$|$\triangle HR(C_2)$|$\triangle HR(C_3)$|All|$D(\mathbf{R}^f)$|
> > |      TrialAttack    |    0.0182|0.0391|0.0127|0.0201|6.17|
> > |       TNA       |    0.0099|0.0359|0.0142|0.0181|5.28       |
> > |RevAdv|0.0163|0.0609|0.0328|0.0353|5.97|
> > |PAPU|0.0216|0.0909|0.0375|0.0459|5.06|
> > |DADA-DIV|$\textbf{0.0405}$|$\textbf{0.1191}$|$\textbf{0.0704}$|$\textbf{0.0745}$| $\textbf{7.09}$|
> >
> >
> > We can observe that under the same dataset, our proposed diversity-aware objective DADA-DIV outperforms baselines on each community and has the largest diversity value. It is because the diversity-aware objective more emphasizes the less affected users in each community when optimizing attackers. Thus, the generated fake user behaviors can manipulate the target RecSys to promote the target item to more users diversely, alleviating the diversity-deficit issue. We have included the ablation study in Tab. 10 in Appx. B.7.

---

> > > ### Author Response · Authors · 2022-08-02
> > > **Response to weakness 2, 3 and 4**
> > >
> > > **CMWp-W2 and CMWp-L2:** The gray-box injective attackers need to access user behaviors as training data. Since this condition looks like a strong assumption, there needs more discussion on training data (real user behaviors) collection. In particular, which types of platforms offer public data to enable the attacks, and whether the collected data is sufficient to train a surrogate model.
> > >
> > > **Reply to CMWp-W2 and CMWp-L2:** Thanks for your suggestions. In the last version, to better simulate real-world applications, we have conducted experiments by taking partial and noisy user behaviors as training data and generating fake user behaviors accordingly. Specifically, in partial data experiments, we set the size of real user behaviors for training as $||\mathbf{R}^p||\_0= \beta\_3  ||\mathbf{R}^r||\_0$, and vary the real user behavior ratio by $\beta\_3 \in \\{1,0.1,0.01,0.001\\}$. In noisy data experiments, we set the size of randomly added noisy user behaviors as $\beta\_4  ||\mathbf{R}^r||\_0$, and vary the added noisy user behavior ratio by $\beta\_4 \in \\{0,0.01,0.1,1\\}$.
> > > As shown in Fig. 4 and 5 in Appx. B.5, the experiments on three datasets show that our proposed attacker outperforms current attackers and can achieve satisfactory performance under a very limited and noisy training dataset, such as a 0.001 ratio of user behaviors, demonstrating the practicability in real-world applications.
> > >
> > > Basically, we can collect user behaviors from social media platforms (e.g., IMDB\[1\], rottentomatoes\[2\], Goodreads\[3\], and Youtube\[4\]) and e-commerce platforms (e.g., Amazon\[5\] and Yelp\[6\]). These platforms allow us to access the item interaction histories of users or the comments of users on items. Then, we can construct user behaviors by crawling the rating histories of users from their profiles and reviews on items. Particularly, even though the collected user behaviors may be incomplete and have some noise, they can be used to generate effective fake user behaviors. It is because they can mimic the real distributions of all users. We have included this discussion in Appx. B.5.
> > >
> > >
> > > \[1\] https://www.imdb.com/
> > >
> > > \[2\] https://www.rottentomatoes.com/
> > >
> > > \[3\] https://www.goodreads.com/
> > >
> > > \[4\] https://www.youtube.com/
> > >
> > > \[5\] https://www.amazon.com.au/
> > >
> > > \[6\] https://www.yelp.com/
> > >
> > >
> > >
> > > **CMWp-W3 and CMWp-Q2**: Given that several stochastic components are involved here, such as the target item selection and k-means initialization, are the reported results aggregated in multiple runs? If experiments are conducted over multiple runs, it is better to state the setting clearly and report the error bars.
> > >
> > > **Reply to CMWp-W3 and CMWp-Q2:** Thanks for your comments. We are sorry that we missed the detailed setting of the motivation experiments and main experiments. We have added the experiment setting in Sec. 4.1.
> > >
> > > The motivational examples (e.g., Fig. 1, 2, and 3) are plotted in one run, which have similar trends. The main experiment results (e.g., Tab. 1, 2, 6, 7) are the average results over three runs. Due to space limit, we do not add error bar. We agree that the error bar is important, which can reflect the stability of attackers. we will add them after summarizing all results.
> > >
> > >
> > > **CMWp-W4:** There are some presentation issues, and the paper should be polished: (1) Phrases like "without loss of generality" and "users contribute more gradients" are used imprecisely. (2) There are some language and grammatical errors.
> > >
> > > **Reply to CMWp-W4:** Thanks for your comments. We have followed your instructions to modify these phrases and polish the writing of the paper by fixing language and grammatical errors.

---

> > > > ### Author Response · Authors · 2022-08-02
> > > > **Response to questions and limitations**
> > > >
> > > > ## Replies to Questions
> > > >
> > > > We have replied **CMWp-Q1** and  **CMWp-Q2** in the reply of  **CMWp-W1** and  **CMWp-W3**, respectively.
> > > >
> > > > **CMWp-Q3:** As an ablation, it is interesting to study some of the existing objectives (e.g., RevAdv) by zeroing out the gradients of users if the target item is in their top-k recommendation list. It can more explicitly reveal the promotion gain from zero-outing the gradients of successfully promoted users and focusing on easier unsuccessfully promoted users.
> > > >
> > > > **Reply to CMWp-Q3:** Thanks for your suggestions. We have added this interesting ablation study in Tab. 9 in Appx. B.7. Specifically, we compare four effective baselines, including TrialAttack, TNA, PAPU, and RevAdv. We set the gradients of a user as 0 if its recommendation list includes the target item, and the loss function $\mathcal{L}\_{base}(\cdot)$ of each baseline can be modified as:
> > > >
> > > > $$
> > > > \mathcal{L}\_{base}(M_\theta,u,t) =
> > > >     \begin{cases}
> > > >             \mathcal{L}\_{base}(M_\theta,u,t), & \qquad t \notin \Gamma_u  \\\\
> > > >         0,\qquad & \qquad t \in \Gamma_u
> > > >     \end{cases}
> > > > $$
> > > >
> > > >  This variant of each baseline is denoted as [baseline]++. Specifically, we employ WRMF as the target RecSys and inject $0.01|U^r|$ fake users. We use HR@50 as the evaluation metric and report results as follows:
> > > > | model | ML-100K |ML-100M|Gowalla|
> > > > | :-----------: | :---------------: | :---------------: | :-------------: |
> > > > |  TrialAttack  | 0.3902|0.1084|0.0105|
> > > > | TrialAttack++ |0.3976 (+1.90%)| 0.1093 (+0.83%)| 0.0107 (+1.90%)|
> > > > | TNA |0.3801|0.1068|0.0109|
> > > > | TNA++|0.3892 (+2.39%)| 0.1076 (+0.75%)|0.0112 (+2.75%)|
> > > > |RevAdv|0.4167|0.1168|0.0121|
> > > > |RevAdv++| 0.4231 (+1.54%)|0.1210 (+3.60%)| 0.0122 (+0.83%)|
> > > > |PAPU|0.4224|0.1126|0.0125|
> > > > |PAPU++| 0.4284 (+1.42%)| 0.1195 (+6.13%)|0.0129 (+3.20%)|
> > > > |DADA-DICT| $\textbf{0.4380}$ | $\textbf{0.1217}$ | $\textbf{0.0134}$ |
> > > >
> > > > As shown in above table, the variant of each baseline outperforms the original baseline on three datasets. It indicates that zeroing out the gradients of users whose recommendation lists contain the target item is beneficial, enabling attackers to generate fake users that affect more users. Moreover, our proposed difficulty-aware attack objective DADA-DICT outperforms all the variants of baselines. It demonstrates that except for zero-outing the gradients of successfully promoted users, it is preferable to concentrate on easy users, as it is easier to promote the target item for them.
> > > >
> > > > **CMWp-Q4:** The description of the attack objectives describes the case where there is one target item. However, in the empirical analysis, there are five items as the target items. When will you zero out the gradients of a user? When one of the target items is in the top-k recommendation list or when all the target items are in recommendation list?
> > > >
> > > > **Reply to CMWp-Q4:** Thanks for pointing it out. We zero out the gradients of a user until all target items are in its top-k recommendation list. Specifically, the combined attack objective on multiple target items $I_t$ is the sum of the attack objectives on each target item $t \in I_t$  in Eq. (6). Then, if one target item $t$ is in the recommendation list of user $u$, we set the attack objective value of $u$ regarding $t$  as 0. We have added this discussion in our experiment setting in Appx. B.2.
> > > >
> > > > **CMWp-Q5:** Why is the difficulty-aware objective called $Ten(\cdot)$?
> > > >
> > > > **Reply to CMWp-Q5:** Thanks for pointing it out. Intuitively, if one user has a higher tendency towards the target item, it is simpler to promote the target item to the user. Thus, we measure the difficulty of users by their tendency score towards the target item. But we have realized that  $Ten(\cdot)$ is not explicit enough to denote the difficulty-aware objective, so we have used the $\underline{di}ffi\underline{c}ul\underline{t}$  abbreviation $Dict(\cdot)$ to denote it.
> > > >
> > > > ## Replies to Limitations
> > > >
> > > > We have replied **CMWp-L2** in the reply of **CMWp-W2**.
> > > >
> > > > **CMWp-L1:** The collected training user behaviors and the fake user budget may be changed. It is better to briefly mention and discuss where there exists a warm-start to create the new fake user behaviors by using previously created fake user behaviors.
> > > >
> > > > **Reply to CMWp-Q5:** Thanks for pointing it out. This is a promising future direction that our attacker and existing attackers do not consider. Thus, we add it as our future direction and discuss it in Appx. B.11.
> > > >
> > > > Here we introduce a fine-tuning warm-up strategy. Basically, when the training dataset changes, we can first train the surrogate RecSys by the new data. Then, we can fine-tune the previously generated fake user behaviors based on the re-trained surrogate RecSys. In addition, when the fake user budget changes, such as when the budget increases, the previously generated behaviors of fake users can be fine-tuned together with the new fake users. Similarly, when the fake user budget decreases, we can employ the partial fake users previously created for fine-tuning.

---

> > > > > ### Comment · Reviewer_CMWp · 2022-08-08
> > > > > **Thank you for the detailed response.**
> > > > >
> > > > > Thanks a lot for the extensive response to the review and for providing the additional clarifications and results! I think they help the paper in terms of clarity and soundness and I have updated my rating accordingly.
> > > > >
> > > > > My main issue regarding the diversity-aware objective still mostly remains. I get what impact the objective has, but I still have issues with the framing. To me, the “diversity-aware” objective seems to be about efficient-awareness, i.e. focussing attack effort on users where it will be most impactful. I believe that this might increase diversity in practice, but that’s more of a (useful) explanation for why this objective works rather than what it appears to be designed to do. From a “by-design” diversity objective I would expect something like clustering the users and then picking users from each cluster or for it to include some sort of regularisation term that tries to incentivise dissimilar users. (Perhaps some of the methods for selecting a diverse set of users developed in the fairness literature might be interesting here). Right now, my issue with calling the the objective “diversity-aware” rather than e.g. “efficiency-aware which happens to increase diversity in practice” is that this framing can be misleading and give readers a false expectation of what the objective optimises for.
> > > > > However, the additional quantitative results regarding the changes in diversity by different attacks seem to indicate that DADA-DIV indeed improves targeting diversity, which helps in alleviating some of the concerns.
> > > > >
> > > > > One suggestion regarding the diversity metric would be to take a look at inequality indices in the economics literature, since they are explicitly designed for such scenarios.

---

> > > > > > ### Author Response · Authors · 2022-08-09
> > > > > > **Response to questions from discussion**
> > > > > >
> > > > > >
> > > > > > We sincerely thank your efforts for helping us make the paper clearer and stronger! To better answer your questions, we summarize the questions from your reply and discuss them as follows.
> > > > > >
> > > > > > **Q1:** (1) Why the diversity-aware objective is not called as the efficiency-aware objective? (2) It is better to add more useful explanations for why the diversity-aware objective works rather than what it appears to be designed to do. (3) From a “by-design” diversity objective, I would expect something like clustering the users and then picking users from each cluster or for it to include some regularisation terms that try to incentivise dissimilar users.
> > > > > >
> > > > > > **Reply to Q1:**
> > > > > > Thanks for your questions.
> > > > > > (1) In the diversity-aware objective, if the real users are less affected by fake user behaviors, we will increase the weight of these real users in the subsequent gradient computation.
> > > > > > Then, the attacker will be optimized to generate fake user behaviors that can affect these users.
> > > > > > In such a way, our attacker can affect more real users diversely. It is the reason we call it diversity-aware. However, we do not increase the efficiency of accelerating the attacker training. It is because we also involve all real users in the attacker parameter optimization as the existing attackers. But we increase the weight of less affected real users in the gradient computation when optimizing attackers, enabling the generated fake user behaviors to affect them.
> > > > > >
> > > > > >
> > > > > > (2) We will add more explanations in Sec. 3.2.1 on why the diversity-aware works and can enable the fake user behaviors to affect the real users more diversely from the perspective of attacker parameters optimization.
> > > > > >
> > > > > > (3) These two approaches are very interesting and promising. Thus, we will add ablation experiments to more comprehensively investigate them in Appx. B.7.
> > > > > >
> > > > > >
> > > > > >
> > > > > > **Q2:** One suggestion regarding the diversity metric would be to take a look at inequality indices in the economics literature, since they are explicitly designed for such scenarios.
> > > > > >
> > > > > > **Reply to Q2:** Thanks for your suggestions. In economics, Gini index \[1\] is widely used to measure the income inequality of a nation or a social group. If the Gini index is lower, the income of people is more equal, i.e., people tend to have similar income.
> > > > > >
> > > > > >
> > > > > >
> > > > > > Inspired by Gini index, we can define a similar metric to qualitatively measure the diversity of affected real users for attackers. Specifically, for each real user $u \in U^r$,
> > > > > > we can compute the tendency difference $\triangle \mathbf{S}\[u\]\[t\]=\mathbf{S}\[u\]\[t\] - \mathbf{S\'}\[u\]\[t\]$ on the target item $t$ after and before injecting fake user behaviors. Then,
> > > > > > we can regard the tendency difference $\triangle \mathbf{S}\[u\]\[t\]$ towards the target item $t$ as the income of user $u$ brought by the generated fake user behaviors. Therefore, similar to the Gini index, we can define the diversity metric as follows:
> > > > > >
> > > > > > $$
> > > > > > GINI\-Div(\mathbf{S};U^r,t,\mathbf{S}') = \frac{\sum_{v \in U^r}{\sum_{u \in U^r}{|\triangle \mathbf{S}\[v\]\[t\]-\triangle \mathbf{S}\[u\]\[t\]|}}}{2|U^r|\sum_{u \in U^r}{\triangle \mathbf{S}\[u\]\[t\]}}
> > > > > > $$
> > > > > >
> > > > > > Lower $GINI\-Div(\cdot)$ implies that the tendency difference of real users tends to be the same. It indicates that attackers can affect real users more diversely. We will use this metric to measure the diversity brought by attackers and add it in the ablation study in Appx. B.7.
> > > > > >
> > > > > > \[1\] https://en.wikipedia.org/wiki/Gini_coefficient

---

### Review · Ethics_Reviewer_NxaS · 2022-08-02

**Recommendation:**

I think the authors have dealt with this issue adequately; it should not affect acceptance of the paper at this point.

**Ethical Issues:**

Yes

**Ethics Review:**

Attack papers have a clear potential negative impact: the attack could, potentially, be used against a production recommender system to harm users or society or fraudulently obtain profit.

Responsible attack research, however, is useful to highlight these potential attacks so that the community can be aware of them and defenses can be developed. I also do not think that defenses are necessarily required for publication - each case should be weighed on its own merits.

In this case, I see the following relevant points:
- This is not a new class of attacks - injection attacks have been well-known for 20-25 years
- The proposed research may make these attacks more effective
- There remains substantial engineering effort required to actually inject the attack profiles into the system in order to attack a deployed system in a way that cannot be detected by its other fraud detectors - these attacks seem unlikely to be usable as-is

---

### Review · Ethics_Reviewer_vDES · 2022-08-06

**Recommendation:**

I recommend the authors add a more in-depth discussion of ethical questions, what has been done to mitigate potential negative implications of this research, and why the benefit of publishing this work now outweighs waiting until defenses are identified or the most vulnerable systems have been notified.

**Ethical Issues:**

Yes

**Ethics Review:**

This paper identifies limitations of current injection attack on recommender systems and proposes two advances to make such attacks more effective.

This is an interesting intellectual contribution.  But there are also important ethical questions to consider in this research.  These questions should be discussed, and the authors' thoughts and decisions clearly stated:
    * Are there any critical applications or actors that are more or most at risk from this attack?  Is there specific guidance for them?  Have they been approached and notified of this attack already?
    * Do all details of the attack need to be published to achieve the benefits of making people aware of this improved attack?  E.g., are there key details that could have been withheld, or is that infeasible?
    * What is the benefit of publishing on the attack before a defense is better understood?
    * Are there further implications of this work that should be considered?

---

### Meta-Review · Area_Chair_cJfQ · 2022-09-05

**Recommendation:** Accept
**Confidence:** Less certain

**Metareview:**

This paper explores an interesting new attack vector against recommender systems -- one that targets recommending an item to users whose top-K lists are easier to manipulate for that item and to a diverse set of users from different interest communities.

Overall, the paper does a reasonable job with showcasing the effectiveness of the attacks. However, there are concerns both about the actual effectiveness of the attacks in practice, where defenses can use external information to detect the fake user profiles (based on their behaviors) used to conduct the attack as well as about the ethical concerns raised by the lack of an appropriate defense strategy for the attack, if it were really effective in practice. The authors are strongly encouraged to discuss the limitations and ethical concerns raised by the attack.

In summary, this is a paper that deserves to be accepted out of procedural fairness -- it has been rated well by the three reviewers. It does describe a cute intellectually interesting attack, even if it is unlikely to be practical or useful.

**Award:**

No

---

### Decision · Program_Chairs · 2022-09-14

Accept